# Disproportionate declines of formerly abundant species underlie insect loss

Roel van Klink[1,2 ✉], Diana E. Bowler[1,3,4,5], Konstantin B. Gongalsky[6], Minghua Shen[1,2], Scott R. Swengel[7] & Jonathan M. Chase[1,2]

Studies have reported widespread declines in terrestrial insect abundances in recent years[1–4], but trends in other biodiversity metrics are less clear-cut[5–7]. Here we examined long-term trends in 923 terrestrial insect assemblages monitored in 106 studies, and found concomitant declines in abundance and species richness. For studies that were resolved to species level (551 sites in 57 studies), we observed a decline in the number of initially abundant species through time, but not in the number of very rare species. At the population level, we found that species that were most abundant at the start of the time series showed the strongest average declines (corrected for regression-to-the-mean effects). Rarer species were, on average, also declining, but these were offset by increases of other species. Our results suggest that the observed decreases in total insect abundance[2] can mostly be explained by widespread declines of formerly abundant species. This counters the common narrative that biodiversity loss is mostly characterized by declines of rare species[8,9]. Although our results suggest that fundamental changes are occurring in insect assemblages, it is important to recognize that they represent only trends from those locations for which sufficient long-term data are available. Nevertheless, given the importance of abundant species in ecosystems[10], their general declines are likely to have broad repercussions for food webs and ecosystem functioning.

Scientific, public and policy-related interest in the plight of insects has soared in recent years[11,12], owing largely to reports of considerable losses of insect abundance and biodiversity[1–4] and fears of concomitant declines in associated ecosystem services[11,13]. Often confused, however, is exactly which measures of insect biodiversity are being considered. Biodiversity is not a single metric, but rather a generalized concept that encompasses the numbers of individuals and species, species' relative abundances (for example, evenness and numbers of rare and common species), as well as the identities of species and their interactions. All of these aspects of biodiversity inform the changes that are occurring to insect biodiversity, and might reveal hitherto-overlooked–but crucial–changes. However, although declines in insect abundance and biomass have been shown in large-scale studies and syntheses[2,5,6], trends in other biodiversity metrics have been less clear. Species richness, for example, has been found to decline along with abundance in some large studies[5,14], whereas in other studies, insect richness was reported to be stable[6,7].

A better understanding of declines in insect abundance can emerge through the study of multiple (complementary) measures of biodiversity[15], and, in particular, by examining trends in rare and abundant species. For example, even if overall abundances of terrestrial insects are declining[2], there are several possible scenarios by which other metrics of biodiversity can concurrently change depending on how species with different relative abundances respond[16]. Box 1 presents

some of these scenarios, particularly focusing on changes in initially abundant and rare species that can strongly influence various biodiversity metrics. We here define 'abundant' and 'rare' species along the continuum of abundances in a local assemblage (that is, where the species falls within the 'species abundance distribution'[17] (SAD)), rather than according to their regional occupancy or range size. Abundant species contribute the most individuals to any given assemblage, and their trends should therefore have the strongest influence on the changes in total abundance[16]. In fact, a decline in total abundance is unlikely without declines of the most abundant species. If we assume that the most abundant species decline on average, three simplified scenarios are conceivable for the rarer species. Relative to abundant species, rare species may, on average, (1) decline proportionally, (2) decline less or (3) decline more. Each of these scenarios gives distinctive signatures in various biodiversity metrics and in the shape of the SAD (see Fig. 1 and Box 1). There are other possible scenarios consistent with total abundance declines–for example, the most abundant species showing no declines–but this would require the unlikely scenario of extinction of most of the rarer species.

Here we examine the multidimensional nature of biodiversity trends in assemblages of terrestrial insects, arachnids (spiders and mites) and Entognatha (springtails and allies)–hereafter described collectively as 'insects', for brevity–during the past decades using a large compilation of insect surveys through time[18]. Although previous work has examined

¹German Centre for Integrative Biodiversity Research (iDiv) Halle-Jena-Leipzig, Leipzig, Germany. ²Department of Computer Science, Martin-Luther-University Halle-Wittenberg, Halle (Saale), Germany. ³Institute of Biodiversity, Friedrich Schiller University Jena, Jena, Germany. ⁴Department of Ecosystem Services, Helmholtz-Centre for Environmental Research (UFZ), Leipzig, Germany. ⁵UK Centre for Ecology & Hydrology, Crowmarsh Gifford, UK. ⁶A.N. Severtsov Institute of Ecology and Evolution, Russian Academy of Sciences, Moscow, Russian Federation. ⁷Unaffiliated: Scott R. Swengel. ✉e-mail: roel.klink@idiv.de

## Box 1

# Conceptual relationships between biodiversity metrics and mean population changes

We start with the assumption of overall abundance declines, on the basis of results from previous studies[2,6], although scenarios could be drawn for other cases. In Fig. 1, we show three conceptual scenarios (see Methods) by which initially abundant and rare species can change through time, illustrated by changes in the population abundances of species (Fig. 1a). These changes, in turn, lead to changes in several diversity metrics (Fig. 1b; insets represent modelled slope estimates, in relation to 0 at the dashed line), as well as changes in the numbers of species in different initial abundance intervals of the SAD (Fig. 1c) and in the population abundance trends of species in these abundance intervals (Fig. 1d). For each scenario illustrated, we start with a simple community of 43 species and 211 individuals with a typical skewed SAD comprising a few abundant species and many rare species[45], and assume similar changes in total abundance through time (Fig. 1b, 'Total abundance').

(1) In the first scenario, all species decline proportionally (that is, the same percentage decline each year) (Fig. 1a, left). In this case, species richness will decline as the rarest species go extinct, whereas Simpson's diversity index (inverted and converted to its effective number of species[28])—which weights abundant species more heavily—does not change, because the relative abundances of all species remain the same. Evenness increases slightly as the species abundances converge (bound between 1 and the highest value; Fig. 1b). Furthermore, the SAD will show a decline in the number of very abundant species, and probably a moderate increase in the number of rare species, as species that are intermediately abundant at first move to lower abundance intervals, but the exact outcome depends on how many rare

species are lost (Fig. 1c). The mean population trends of all initial abundance intervals will be the same (Fig. 1e).

(2) In the second scenario, the abundant species decline more than the rare species do (Fig. 1a middle). In this scenario, species richness does not change, because no rare species are lost, but diversity and evenness increase through time as the relative abundances become more similar (Fig. 1b). Because of these changes, the SAD shows a strong increase in the number of rare species, as species that were initially abundant move to the interval with those already rare (Fig. 1c). Finally, at the population level, initially abundant species have more negative population trends than do initially rare ones (Fig. 1d).

(3) In the third scenario, the rare species decline more than the abundant species do (Fig. 1a right). In this scenario, species richness declines more strongly than in the other scenarios, as more species go extinct, and diversity and evenness also decline (Fig. 1b). Here, there will be a decline in the number of rare species in the SAD, because the abundant species do not decline so much that they become rare and do not compensate for the loss of rare species (Fig. 1c). At the population level, species in the lowest abundance intervals show the strongest declines (Fig. 1d; the deviation from the straight line for the lowest abundance interval is caused by rounding to integers[31]).

Note that in these simplified scenarios we did not include colonization by new or invasive species, or increasing populations. Colonists and increasing species could balance out some declining species. The code for these simulations is provided in ref. 31.

some of these multifaceted patterns for subsets of taxa (for example, butterflies[14] or hoverflies[16]), locations[6,7] and/or habitat types[19], our synthetic approach allows us to assess the prevailing trends in multiple biodiversity metrics from the openly available long-term insect assemblage data across locations, taxa and habitats. Our focus here is on terrestrial insects only, because two recent studies from Europe[20] and North America[21] have used extensive data compilations to assess changes to the diversity of freshwater invertebrates. We identified 89 terrestrial studies from our previous data compilation[16] in which one or more metrics of biodiversity either were provided or could be calculated from the provided data (see Methods). We added newly published years to the existing data, and fixed some minor data issues (updates available in the online repository[22]). To this, we added 17 new studies from the literature published between 2018 (the year of compilation by ref. 18) and 2021. We also searched the literature for studies in Spanish, Portuguese and Chinese to complement our original search in English and Russian, which allowed us to add two more studies. In all, we were able to analyse data from 106 studies with 923 total sampling sites (Extended Data Fig. 1 and Supplementary Tables 1 and 2), spanning between 9 and 64 years. Of these, 75 studies provided measures of at least two metrics of biodiversity (usually abundance and richness) and 57 studies (from 551 sites) provided full community data (that is, all taxa resolved to a consistent taxonomic level, usually species level) so that we could consider trends in multiple metrics of biodiversity simultaneously. We analysed the trends in total abundance (largely the same analyses as described previously[2], but with the newly added studies), several metrics of diversity (for example, species richness,

Simpson diversity), a metric of evenness, and trajectories of populations of species within the assemblage. We used hierarchical Bayesian autoregressive models to account for study- and site-level variation. Our models present the yearly change estimates and the 80%, 90% and 95% credible intervals (CI) of these estimates. Following previous studies[2,23], we interpreted any 95% CI that did not overlap zero as strong evidence for a directional trend (that is, we are more than 97.5% certain that there is a directional trend in our data), whereas a CI overlapping zero between the 90 and 95% CIs was interpreted as moderate evidence, and an overlap between the 90% and 80% CIs was interpreted as weak evidence (that is, a 10% chance that the actual mean of the data is zero).

As expected from our previous analysis[2], we found strong evidence for an overall decline in the total abundance of insects in this updated and expanded dataset of terrestrial assemblages. The overall change in abundance was −1.49% per year (95% CI: −2.18, −0.79) (Fig. 2), but there was considerable variation at the dataset level (interquartile range of the dataset-level predictions: −2.28% to −0.53%; Extended Data Fig. 1). This general trend was similar but slightly weaker (−1.22% (CI: −2.01, −0.42)) when we restricted the analyses to datasets that provided full community data (n = 57 studies; dotted lines in Fig. 2). We next determined whether these abundance declines translated into changes in other metrics of biodiversity, and, if so, which scenarios from Box 1 were most or least consistent with those changes. We found moderate evidence for an overall decline in species richness across all assemblages (−0.29% annually (95% CI: −0.64, 0.06); −0.20% (−0.57, 0.15) on the full community dataset; Fig. 2). This negative overall

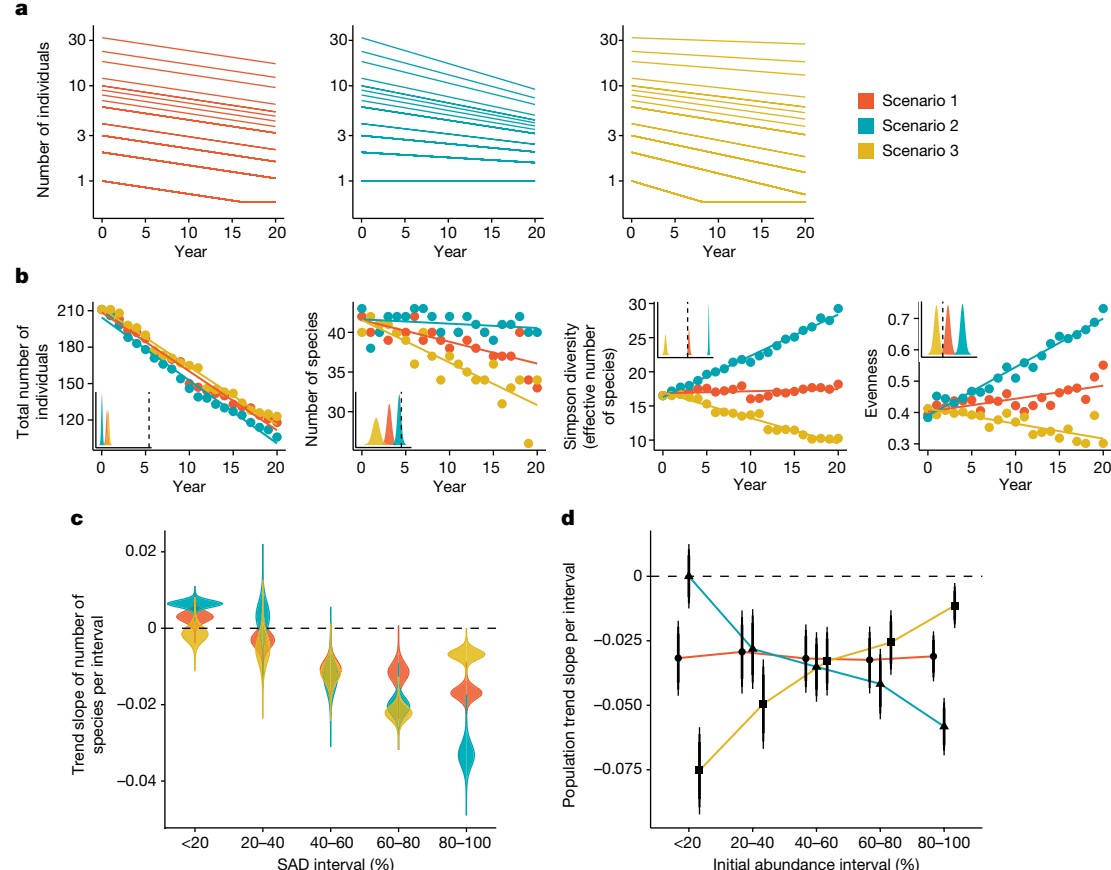

**Fig. 1 | The three conceptual scenarios described in Box 1 give rise to distinct changes in biodiversity patterns. a–d,** Effects of three conceptual scenarios by which species change over time (all species decline in proportion to each other (scenario 1, red); abundant species decline more than rare species do (scenario 2, turquoise); or rare species decline more than abundant species do (scenario 3; gold)) on population abundances (**a**), four biodiversity metrics (**b**), the numbers of species in different SAD intervals (**c**) and the mean population abundance trends of species in these abundance intervals (**d**). For biodiversity metrics that depend on species counts (species richness and evenness (**b**) and species richness per SAD interval (**c**)), a small amount of error was added. The insets in **b** represent modelled slope estimates in relation to 0 at the dashed line. Simpson diversity was converted to its effective number of species[28]. See 'Extraction and calculation of biodiversity metrics' in the Methods for an explanation of the biodiversity metrics. Note that because this simple conceptual model is intended to mimic a real dataset to some extent, the trends should be interpreted qualitatively rather than quantitively.

trend in species richness, albeit slight, contrasts with several analyses of insects and other taxa that found little evidence for consistent directional trends[6,7,24].

Although concomitant declines in abundance and species richness are expected from a sampling process (that is, fewer species would be expected when fewer individuals are sampled from a species pool[15,25]), species' relative abundances might also change. We found no evidence for any trends in diversity on the basis of individual- and coverage-based rarefaction[26,27], nor in Shannon's and Simpson's diversity metrics (converted to their effective number of species[28]), which differentially weight abundant species (Fig. 2), suggesting no changes in relative abundances. Sensitivity tests showed that these trends were robust to the exclusion of very long and very short time series (Extended Data Fig. 2). However, when we restricted our analyses to studies that included many sites (10, 20, or 50), we found weaker declines in abundance and richness (Extended Data Fig. 3).

Our finding of a lack of temporal trends for these diversity metrics could imply that there was no change in the relative abundances of species through time[15,25]. However, we did find strong evidence for an increase in a metric thought to reflect evenness[29], calculated as the ratio of the inverse Simpson index to the number of species (Fig. 2). Because this metric is calculated by dividing changes in species richness (which was declining) by inverse Simpson diversity (which was not changing), an increase could be expected. However, because our observations

of changes in species richness appear to be driven largely by changes in the total abundance of insects (as indicated by the lack of trends in rarefied richness), changes in this metric need not reflect changes in relative species abundances.

On the basis of the patterns of the different biodiversity metrics (Fig. 2), a scenario of steeper declines of rare species is highly unlikely (scenario 3, Box 1). However, these patterns could be consistent with the proportional declines in scenario 1 (for example, a lack of change in the diversity indices), or with the steeper declines of initially abundant species (increasing evenness) in scenario 2.

To distinguish between these scenarios, we compared trends in the number of species in different abundance categories (as in Fig. 1d). For this, we first divided the total SAD of each site into five equally sized intervals of abundance, and then calculated the number of species in each interval each year (for a visual explanation, see Extended Data Fig. 4). We found the strongest declines (−0.80% annually (−0.96, −0.65%)) in the number of species in the highest abundance interval (Fig. 3), but there was also moderate to strong evidence for declines of species in the intermediate abundance intervals (between −0.33% and −0.43% annually). The lowest abundance interval (0–20%) showed little evidence for any trend. This interval contains the highest number of species (Extended Data Fig. 4), and represents several categories of trends, such as stable rare species, species that were previously more abundant, species that went locally extinct and newly colonizing

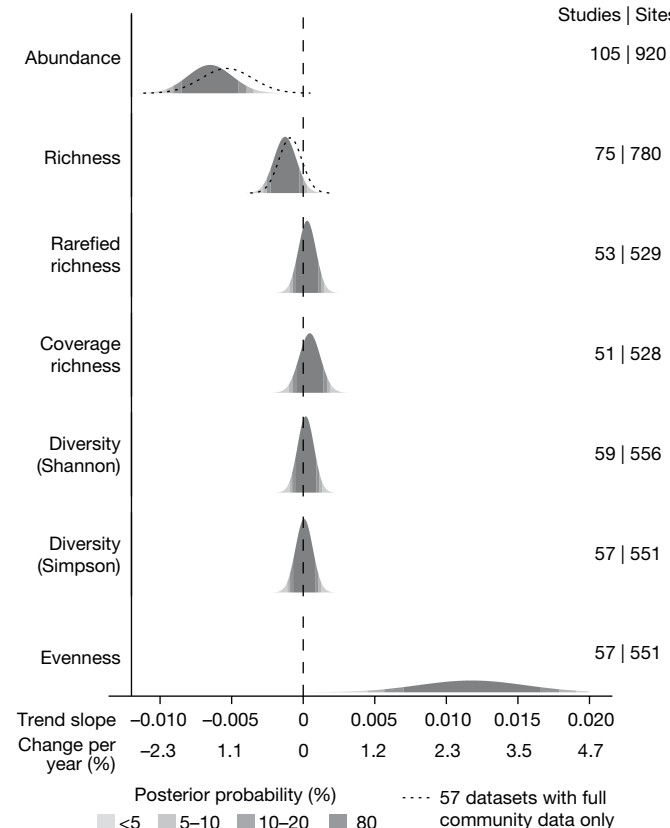

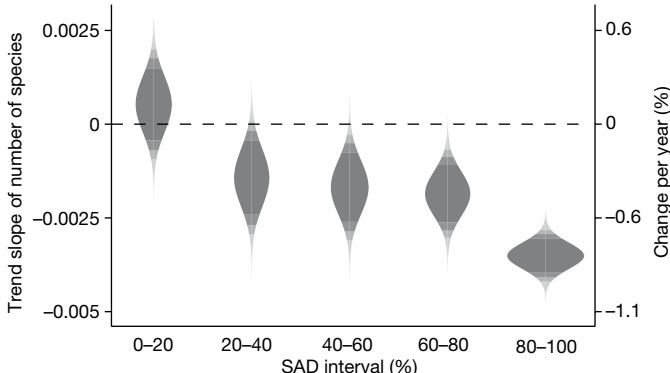

**Fig. 3 | Probability density of the trend estimates for the number of species in five abundance intervals over time.** The five ($\log_{10}$-transformed) SAD intervals were assigned separately for each time series, scaled to the highest abundance of any taxon observed in any year. Shading as in Fig. 2.

**Fig. 2 | Probability density of the posterior trend estimates of the changes in univariate biodiversity metrics.** Shading represents the posterior probability of the mean slope estimate, corresponding to the 80%, 90% and 95% CIs from the hierarchical Bayesian models. Numbers indicate the numbers of studies and sites available for each metric. Abundance is the total number of individuals observed at each time point, richness is the number of taxa observed at each time point, rarefied richness represents the expected number of species to be seen for the minimum number of individuals observed in any year in the time series, coverage richness is the expected number of species if 80% of the assemblage had been surveyed, Shannon and Simpson diversity were converted to their effective number of species and evenness was calculated as the ratio between the inverse Simpson index and species richness. The dotted lines for abundance and richness represent the results based on only the 57 studies with full community data.

species. We found similar results using alternative ways to bin species (see Extended Data Fig. 5). These patterns are again consistent with either the proportional declines scenario, or the scenario of stronger declines for abundant species, but not with the scenario of greater declines of locally rare species from Box 1.

Declines in the number of species in the higher abundance intervals might give support to the scenario of disproportionate declines for initially abundant species. However, proportionate declines are still possible, because this would also result in the greatest loss of species from the highest abundance interval, as losses from lower abundance intervals are partially offset by gains of species that were formerly in higher abundance intervals (Box 1). To disentangle these possibilities, we tested whether differences in trends were detectable at the population level by calculating the mean population trends of all species within each initial SAD interval. A common pitfall when relating population trends to their initial values is detecting false trends owing to a regression-to the-mean effect[30]. That is, if, for example, an extremely high value is found at the start of a time series owing to stochasticity in population dynamics or sampling, the values for the following years are likely to be lower, which would lead to the false detection of

a declining slope over time. Hence, the species with highest starting values are likely to show the strongest declines, and species with the lowest starting values the strongest increases, simply by returning to their long-term averages. To avoid this, we corrected the detected trends for each abundance interval by adding an expected regression-to-the-mean effect calculated from the longest and most robust time series (see Methods) and also conducted a series of sensitivity analyses (Extended Data Figs. 6 and 7).

Our population-level analyses indicated that species that were most abundant at the start of the time series showed, on average, the strongest declines (−7.72% annually; Fig. 4), despite considerable variability. Species in all other initial abundance intervals also on average showed strong evidence for declines, but with smaller magnitudes (between −4.63% and −6.14% annually) (Fig. 4; for the results of other ways of classifying species by their initial abundances, see Extended Data Fig. 7). We found that the most abundant species still showed the strongest declines when we took the most conservative approach to defining abundance intervals (based on species' mean abundances across the whole time series, hence avoiding regression-to-the-mean effects; Extended Data Fig. 7).

Among the populations with the steepest declines were species from across the insect tree of life (for example, beetles, moths and grasshoppers), and they included both agricultural pests, such as the corn aphid *Rhopalosiphum maidis*, and beneficial insects, such as predatory beetles (Supplementary Data 3 in the code repository[31]). Indeed, such declines in the populations of formerly abundant insect species have been reported from several locations, and include monarch butterflies (*Danaus plexippus*)[32] and other butterfly species[33], the meadow spittlebug (*Philaenus spumarius*)[34] and the now-extinct Rocky Mountain grasshopper (*Melanoplus spretus*)[35]. This potentially common phenomenon of strong declines in formerly abundant species could offer an explanation for the observations that there are generally fewer insects in terrestrial systems than there used to be[2]. To test whether there is indeed a strong association between the mean population trends and the total abundance trends in a dataset, we related the dataset-level random effects of these metrics for all initial abundance groups. Although we found that all abundance groups had strong associations with total abundance loss, the strongest associations were for the more abundant species (Extended Data Fig. 8), supporting our assertion that losses of abundant species underlie the overall declines in insect abundance. Overall, on the basis of our compilation of openly available insect time-series data, we find that changes in local insect biodiversity are dominated by losses in total abundance and weaker losses of species richness, with the strongest declines observed for species that were more abundant at the start of the time series. Of the

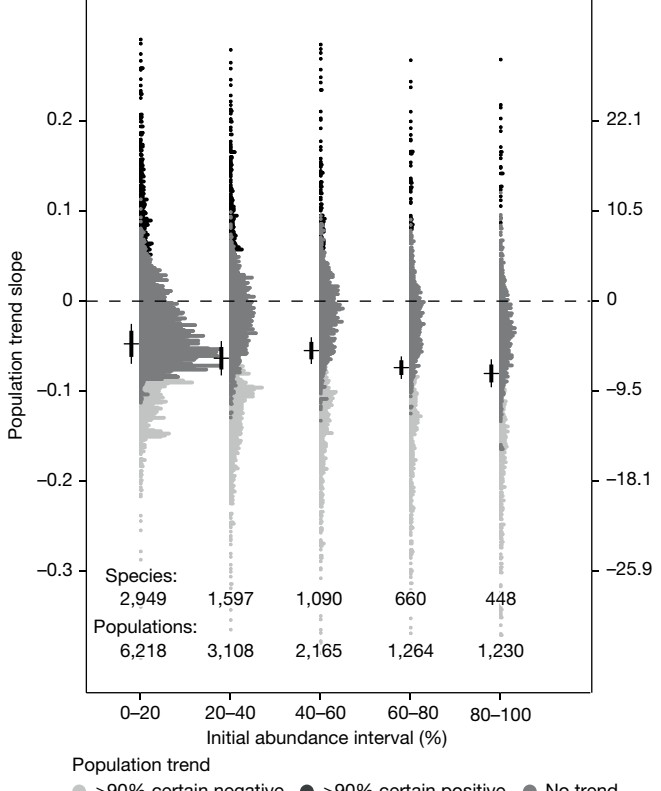

**Fig. 4 | Population trend estimates per initial abundance interval, corrected for regression-to-the-mean effects.** Black horizontal bars and error bars represent the mean, 80% and 95% CIs of the mean estimates. The dots represent the population-level random effects, coloured by trend direction and certainty. The most extreme values are not shown, to aid visualization.

scenarios described in Box 1, these findings are most in line with scenario 2 (stronger declines for formerly abundant species), through stronger trends of taxa with a higher initial abundance; they also have some characteristics of scenario 1 (proportional declines), with stable diversity metrics and declines in richness. We argue that the most likely explanation for these patterns lies somewhere in between the two: initially abundant species decline strongly, but proportionally to each another, whereas the declining rarer species are often—but not always—replaced by other (now) rare species, leading to declines in richness. This result counters studies showing that biodiversity change is characterized by common species faring better (or less badly) than rarer species[8,9]. These studies used range size, not local abundance, as a measure of rarity, and did not analyse changes in species abundances. Still, reconciling these contrasting findings will require further work.

Despite these overall trends, we must clearly identify the limitations of data availability for our study. Our data, like those in most syntheses, are not a representative global sample, but have a strong bias towards Europe and North America (Extended Data Fig. 1 and Supplementary Table 2). When analysed without data from Europe and North America, we found no trend in abundance or richness (Extended Data Fig. 9), which probably reflects the limited data available and the high variability of trends in other regions, rather than an absence of change. Hence, our estimated mean trends cannot be geographically extrapolated; instead, they simply represent the most up-to-date state of the available knowledge one can gain from existing (open) data.

Although our work here focuses on trends of terrestrial assemblages, which have so far dominated the discussion on insect declines, trends of freshwater assemblages are similarly important. Our previous analysis indicated a widespread increase in the abundance of freshwater insects[2], which was confirmed in a more extensive analysis for European streams that also revealed a mean increase in species richness[20]. By contrast, a recent synthesis of stream ecosystems in the USA[21] indicated strong declines in the abundance but increases in the richness of freshwater insects during the past decades. Much still needs to be done to compare and contrast the trends of different geographical regions and to understand the differences in the changes of terrestrial and freshwater insect assemblages.

Our main finding—the disproportionate declines of initially abundant insect species—could help to explain the frequent observations that there are fewer insects now than in the past. Given the nature of our synthetic analysis across many taxa, systems and locations, we can only speculate on the underlying causes, which are likely to be associated with recent anthropogenic changes. For example, case studies have attributed the declines of some abundant species to climate change[34,36], land-use intensification[37] and decreases in plant nutritional quality[38]. Some species might also have been abundant in the past because they benefited from certain types of historic land use (for example, traditional, low-input agriculture), but have declined more recently as land uses have changed[37,39]. Abundant species are often disproportionately important for ecosystem structure[10], functioning[40,41] and services[42,43], as well as for the diversity and abundance of higher trophic levels[10,36,44], so their declines are likely to have already led to a broad-scale rewiring of ecosystems, and will continue to do so.

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

## Methods

### Data collection and selection

The basis for the current analysis was the InsectChange database[18], which we developed to compile openly available long-term (ten or more years; nine or more years for datasets from under-represented regions) standardized surveys of insect, springtail and arachnid groups (families or assemblages). For the current analysis, we updated the database, which initially focused on assemblage-level abundance and biomass, by extracting species richness and other diversity metrics from these same publications, and including new data that met our criteria published until February 2021. This information included both data added to open-access repositories since our last download, and data from newly published studies. For a new study to be included in our compilation, we required that studies provided two or more calculated biodiversity metrics, over a time span of at least ten years, or the full set of taxa and their abundances for each time point so that we could compute multiple metrics. We searched for new publications fitting these criteria by searching Web of Science (for English language) and https://www.elibrary.ru (for Russian language), using the same search terms that were used to compile the InsectChange database[18] (see Supplementary Methods for the exact search strings). For this study, we increased our global coverage by searching papers published in Spanish and Portuguese using https://www.scielo.org/ and in Chinese using https://www.cnki.net/. This yielded two additional studies with appropriate data. In addition, we searched all papers citing references[1,2]. We also used cross referencing and personal networks to find data. In total, we were able to add 17 new studies from which we could extract data to the InsectChange database, and we updated nine studies that provided more recently published years of data.

For the purposes of our analyses here, we only included studies of terrestrial insects because many of our datasets on aquatic insects within the InsectChange database[18] did not have a consistent taxonomic resolution over time to calculate reliable biodiversity metrics, and larger, more comprehensive syntheses of freshwater insects have recently been published[20,21]. We also excluded datasets that had only biomass measurements, as well as experimental sites at which researchers manipulated the environmental conditions. Within each sample, we excluded invertebrates that did not belong to the classes Insecta, Entognatha (springtails and allies) or Arachnida. For studies in which the provided total abundance or richness values might have included non-insects, we checked that these assemblages consisted of at least 90% insects, arachnids or springtails and excluded them if that proportion was unknown or smaller. Finally, we excluded one study (DatasourceID = 70) that was originally derived from the Global Population Dynamics Database[46] and later included in BioTIME[47] because its methods and methodological consistency could not be verified (M. Dornelas, personal communication).

### Dataset description

In total, for the analysis presented here, the database consists of 106 studies, 75 of which provided at least 2 diversity metrics, and 57 provided raw data from which we could calculate all biodiversity metrics (Supplementary Table 1). The data were heavily skewed towards Europe and North America (69% of studies providing abundance data, and 74% of studies providing compositional data; Supplementary Table 2). The median duration of the studies was 20 years, with a median of 11 sampling years. The number of sites per study ranged between 1 and 138, with a median of 3. More details on the dataset characteristics are provided in Supplementary Table 3, with a full account of all studies[48–143] included in Supplementary Table 4. All data used are available as Supplementary Data 1 and 2 in the code repository[31].

### Extraction and calculation of biodiversity metrics

**Metrics of biodiversity change.** For datasets that provided derived metrics of species abundances or different metrics of species diversity (for example, species richness), we simply included those values in our analyses after first ensuring that investigators performed appropriate standardization in their sampling methodology, or after the fact, in their calculation of metrics. For datasets in which the numbers of individuals of each species were provided, we calculated several biodiversity metrics after ensuring appropriate standardization of sampling (for example, sampling effort, time of year; further discussed below).

For each study, we calculated as many of the following metrics as possible given the data:
- Abundance: Total number of individuals observed per year.
- Species richness: Number of species, morphospecies or higher taxa observed in each year.
- Rarefied richness: Richness accounting for differences in numbers of individuals; that is, the expected number of species in each year, had the number of individuals caught been equal to the lowest total abundance observed in any year in the time series (if the lowest number was lower than ten individuals, we calculated the rarefied richness for ten individuals).
- Coverage-based richness: Expected number of species if 80% of the community had been sampled[26].
- Effective numbers of species given Shannon diversity ($H'$): A diversity metric that quantifies the uncertainty (entropy) in the prediction of species' identity converted to its effective number of species ($e^{H'}$) (ref. 28).
- Effective number of species given Simpson diversity ($D$): A diversity metric that measures the probability of intraspecific encounter (with replacement) converted to its effective number of species: $1/D$ (ref. 28).
- Evenness: The ratio between the effective number of species given Simpson diversity (see above) and species richness $(1/D)/S$. This metric of evenness is among those recommended previously[29] ($I_{1/D}$).

We calculated coverage-based richness using the iNEXT[144] and all other biodiversity metrics using the vegan[145] package for R.

**Changes in the SAD.** To investigate changes in the number of species that were differentially common or rare in the community, we studied changes in the SAD. The SAD describes the distribution of species' relative abundances in an assemblage[17], here quantified through bins of $\log_{10}$-transformed abundances plotted on the $x$ axis, and the number of species in each bin on the $y$ axis. Although the precise shape of this distribution and what this means for communities has been discussed at length[146], we were only interested in changes in the number of species in different SAD intervals.

On a per-site basis, we defined intervals of the SAD and counted the number of species falling into them as follows. First, we created five equally sized intervals of the $\log_{10}$ abundances (Extended Data Fig. 4b), which ranged between zero and the maximum abundance observed at that site over the course of the time series. In other words, the baseline for 'locally abundant' at a site is the highest abundance of any species recorded from the site over the course of the entire time series. Our second way of slicing up the SAD was to assign four quartiles based on all abundance values in the time series (Extended Data Fig. 4c), each quartile having (approximately) the same number of observations (some variation exists because of rounding). Given the skewed SADs typical for insect assemblages[45], the fourth quartile of our second approach is naturally the widest, and can occupy over 50% of the abundance range (Extended Data Fig. 4c). By contrast, with our first approach, the uppermost quartile contained the lowest number of species, composed of the small number of very common species. For each year, we then calculated the number of species whose $\log_{10}$ abundances fell in each

interval or quartile. For ties, we took the number of species smaller than the quartile value. We present the results of the first approach in the main text and those of the second approach in the Extended Data Fig. 5, as both lead to similar conclusions.

**Mean population changes.** To test whether species that were at first more abundant showed more negative temporal trends, we modelled the mean population trend of all taxa in relation to their initial abundances. For this, we used 34,317 populations at 584 sites from 56 studies. We grouped species per site into 6 initial abundance intervals: 5 groups based on the range of $log_{10}$ abundances of the species found at a site in the classification year(s) (similar to the division for the SAD brackets above, but described in more detail in the next paragraph), and a group for the species that were not detected in the classification years. Because we were interested in local rarity, we allocated species by their initial abundances per site. An alternative way of studying trends in relation to species' rarity would have been to look at regional abundance or occupancy, but this was not possible for most of our datasets, For an analysis studying occupancy changes including part of the presently used data, see ref. 8.

To allocate species at each site to an initial abundance interval, we created 5 equally sized intervals between 0 and the $log_{10}$-transformed highest observed abundance value at the start of the time series. We allocated each species to one of the five intervals according to its initial abundance. The species that were absent in the classification year(s) were not analysed because their mean trends would in most cases be positive, leading to a positive mean trend. In the main text, we used the data collected in the first sampling year to define species' initial abundances, but other ways of allocating species by their initial abundance values are possible; for example, on the basis of their average abundance values over years 1 and 2 or years 1 to 5, or their average abundance across the whole time series (that is, assigning generally abundant and rare species, as has been done previously[147]). The effect of such choices on the slope estimates is shown in Extended Data Fig. 7.

We discarded sites with no or only one species detected in the classification year(s), or with only a small number of equally abundant (usually singleton) species (that is, when no vector of relative abundance values could be calculated without infinitely large numbers). In all, between 5 and 25 sites in 10 studies were discarded, depending on the number of years used for classification.

When relating trend slopes to their initial values, one is likely to find the steepest slopes for the species with the most extreme (highest and lowest) starting values, owing to a regression-to-the-mean (RtM) effect[30]. This occurs when, owing to stochastic variation in population dynamics, or random sampling effects, a high value that is drawn in the first year is followed by lower values in subsequent years, closer to the long-term average, leading to an overall apparent negative trend, even when no such decline exists. This can be dealt with by calculating an expected RtM effect from a control group, and adding this to the estimate of a treatment group[148]. Because an independent control group for real-world monitoring data, without stochasticity or sampling variation, is not available, we used the best sampled sites in our dataset to estimate an expected RtM effect. We first selected only those sites with at least 15 years of sampling data (260 sites from 26 datasets), and modelled the mean population trends of each initial abundance group. To estimate the potential contribution of RtM to estimated long-term trends, we separated the data used to estimate initial abundance from the data used to estimate the long-term trends. Thus, we left-censored the data by one year (and three years as a sensitivity test), and modelled the population trends again, while using the censored year to classify the initial abundance. The difference in year slope estimates between the full time series and the censored time series was, for each initial abundance group, used as a correction factor, and added to the estimates of the full analysis, to shrink the final

estimates towards zero. In the main text, we present the corrected data based on one-year censoring; the potential effects of more censoring and the number of years used for classification can be found in Extended Data Figs. 6 and 7 and the calculated correction factors can be found in Supplementary Table 5.

Note that our most conservative method for assigning species to abundance intervals—on the basis of their mean abundances across all years—is not affected by RtM effects. Qualitatively, our results remain the same across these sensitivity analyses.

For datasets that needed sample-based rarefaction to equalize sampling effort over years (see below), we used the rounded median values per species per year as input for these population models.

## Extraction of biodiversity metrics from publications

We extracted pre-calculated biodiversity data from studies that did not provide raw data. We could extract the following metrics across the time series: total abundance ($n = 89$ studies), number of species ($n = 28$), rarefied richness ($n = 3$), Shannon diversity ($n = 4$) and Simpson diversity) ($n = 1$).

Although we endeavoured to run our models on one value per year (to avoid complications from species' phenologies), this was not always possible, because some studies provided these metrics per month or per season. In these cases, any potential seasonal patterns were dealt with by including a random intercept on time period of sampling (see 'Statistical analysis').

**Preparation of raw data.** For studying changes in biodiversity metrics, it is essential that sampling effort over the years is equal. Sampling methods were standardized within sites (for example, type of trap used or type of survey), but could vary in sampling effort (for example, duration of sampling period or number of subsamples). To equalize this variation, we processed all studies that provided raw community data in such a way that each year in each site provided one value for each biodiversity metric, on the basis of the same sampling effort (that is, same number of weeks sampled, same number of traps active and/or same number of subsamples taken), summed over the same season(s). In this way, any phenological differences or shifts among years in sampling effort should be accounted for in the best possible way, and stochastic variation should be minimized. Because every study had a different design, we had to decide on the standardization strategy on a case-by-case basis.

The community data available to us can be divided into three types, with different processing requirements:

(1) The study provided one species-by site-by-year matrix. We used this directly to calculate all biodiversity metrics for analysis.

(2) Species-by-site by-year data were provided together with an account of the sampling effort (number of samples underlying each row is reported, but no raw data were provided per individual sample). Here, we subsampled the years with higher trapping effort to provide the expected community composition if the number of samples had been the same as in the year with the lowest sample size. We repeated this subsampling procedure 100 times, and calculated the means of each biodiversity metric, to be used as model input. In some cases, it was clear that subsampling to the lowest sample size would lead to a large loss of data, and in such cases, it was more economical to discard a year than lose a large amount of data. A practical example: if year 1 has 300 individuals in 6 traps and year 10 has 500 individuals in 4 traps, we subsampled 200 individuals (4/6 of 300) from year 1, so that each year has the number of individuals equivalent to 4 traps. We repeated this resampling 100 times to derive 100 hypothetical community compositions, and calculated the biodiversity metrics and population abundances for each iteration.

(3) Exact data were provided per sampling event (number of individuals per species per trap/site per date), but the number of traps, dates

or periods sampled is variable among years (owing to trap malfunctions or changes in sampling design or sampling frequency). Here, we first divided the years into monthly or bimonthly periods, and then decided whether we could maximize data use by including more months or more years (that is, whether to exclude periods that were not sampled each year, or exclude years that did not have samples in each period). The aim here was to maximize data use, so that we could attain the highest quality of data for a maximum number of years. For each period, we then subsampled the minimum number of samples observed in any year from the available samples, and added these together to form one yearly sample. This was repeated 100 times, and the biodiversity metrics were calculated each time. The mean values of these 100 iterations were used for model input. A practical example: ten pitfall traps per site were usually emptied weekly, but sometimes fortnightly, the trapping period changed from year-round at the start to May–September in later years, and sometimes samples were missing owing to flooding or trampling by livestock. Here, we first discarded all months other than May–September. Then we allocated five monthly periods and calculated the number of sampling days (sum of sampling period × number of traps) for each month. We then drew random samples (without replacement) from the available samples per month, until the total number of sampling days added up to the minimum number sampling days observed. We added the months up to form one yearly sample and calculated our metrics. This was repeated 100 times.

The sampling effort is thus equal among all years in each site, but not necessarily among different sites within a study.

**Taxonomic considerations.** Insects make up the largest class of eukaryotic organisms, but at least 80% of species remain undescribed. In addition, the expertise required to identify insects to species level is high. No person can identify all species in an area, and there are often life stages in which a species can't morphologically be distinguished from other species (for example, juveniles or one of the sexes).

It is therefore no surprise that, in our amassed data, there is a large variability in taxonomic precision among studies, and that there is a trade-off between taxonomic breadth (the number of insect groups assessed) and taxonomic precision (the taxonomic level at which all organisms are classified). To avoid biasing our analysis to those taxa in those regions that can reliably be identified to species level, we took a two-pronged approach:

(1) For studies in which almost all individuals were identified to species level (85% of studies), we avoided artificially increasing our biodiversity-change metrics owing to a small number of imprecise identifications (usually genus or family level). To do this, we developed a simple algorithm to clean the taxonomy: We probabilistically allocated individuals at genus level to one of the observed species in the same genus. If no congenerics were present in the time series, these genera were treated as species. This cleaning was always performed just before calculating the biodiversity metrics, also in randomization models.

(2) For studies that were not identified to species level (15% of studies), we accepted this taxonomic imprecision, and assumed that the data collectors identified all organisms to the best of their abilities. We checked that the taxonomic precision had remained stable over time. The use of taxonomic groupings other than species as a proxy for diversity changes is justified, because various studies have shown that identification to higher taxonomic groupings, or morphologically distinct taxa (morphospecies), is generally sufficient to detect differences in richness and composition over environmental gradients[149]. Of the 34,317 populations, 84.1% were identified to species level, 6.9% to genus, 5.1% to family, 2.7% to suborder and less than 1% to higher taxonomic levels.

## Statistical analysis

To test for temporal changes in these biodiversity metrics, we used autoregressive hierarchical Bayesian models. The fixed model structure was simple, with the biodiversity metric of choice as the response variable, and 'Year' as the only fixed independent variable.

By focusing only on the temporal slopes, we could account for the different scales of measurement (from one to thousands of individuals, and up to hundreds of species per year). The coefficient of the 'Year' variable (the temporal slope) can be back-transformed to the percentage change per year, and we report both in all graphs.

The response data of all biodiversity models with positive exact, estimated or averaged counts (total abundance, species richness, rarefied richness, coverage richness, Shannon diversity, inverse Simpson diversity and the number of species in each of the SAD quantiles) were $\log_{10}(N+1)$-transformed for analysis, and the error structure was assumed to follow a Gaussian distribution. The evenness values were bounded between 0 and 1, and the error structure followed a beta distribution.

We accounted for the non-independence of the repeated measurements at each site and the expected autocorrelation among sites part of the same study by including a series of nested random intercepts and slopes:

We included random intercepts for study; study area (in cases when sites were clustered in different study areas of the same study); site (the smallest reported sampling unit); and within-year time period (finest resolution: month; when samples were collected repeatedly within year, nested within study). We included random slopes for the effect of year (at the levels of study, study area and site), and for temporal autocorrelation by adding an autoregressive term of order 1 (AR1) (as a continuous Ornstein–Uhlenbeck process), on which we placed a random effect at the site-level to allow site-level variation in the strength of autocorrelation.

We fitted these models using integrated nested Laplace approximation (INLA)[150] in R4.2.2 (ref. 151), a Bayesian method that efficiently and accurately approximates Bayesian posterior distributions, without using Markov chain Monte Carlo methods, and which allows for complex layered random effects, including autoregressive terms.

As priors, we used penalized complexity (pc) priors with as hyperparameters a sigma of 3 × s.d. of the data and $\alpha = 0.01$, meaning that there is a probability of 0.01 that the mean lies outside 3 standard deviations of the data.

The prior for the AR1 correlation is defined by INLA on the logit lag one correlation scale (where $\rho$ is the correlation coefficient) and was given a pc prior with a sigma of 0.5 and an $\alpha$ value of 0.01 (probability of 0.01 that the mean is larger or smaller than 0.5).

The general model structure in INLA annotation is:

```
inla(x ~ Year+
f(Period, model='iid', hyper = prior.prec))+ # random intercept
season
f(Plot_ID, model='iid', hyper = prior.prec))+ # random intercept site
f(StudyArea_ID, model='iid', hyper = prior.prec))+ # random intercept
area
f(Datasource_ID, model='iid', hyper = prior.prec))+ # random intercept
study
f(Plot_ID_slope, iYear, model='iid', hyper = prior.prec))+ # random
slope site
f(StudyArea_ID_slope, iYear, model='iid', hyper = prior.prec))+#
random slope area
f(Datasource_ID_slope, iYear, model='iid', hyper = prior.prec))+#
random slope study
f(iYear, model='ou', replicate=as.numeric(Plot_ID_4INLA),
hyper = list(theta1 = list(prior='pc.prec', param=c(0.5, 0.01))))# AR1
term
family="gaussian" # normal distribution for the log-transformed
count data
```

where $x = \log_{10}$ (metric + 1) for the univariate metrics, and untransformed beta diversity values for the turnover models.

**Models of population change.** To estimate the mean population trends of all initial abundance intervals, we adjusted model structure for the aggregate biodiversity metrics above, by adding an additional layer of random intercepts and slopes (for unique species × site combinations), allowing random variation among species. The autoregressive term was retained at the site level, because the population-level autoregression required excessive memory use (more than 1 TB) and never converged when included at the species × site level. Here, we used a Poisson error structure (that is, no log transformation) to allow for zero abundances. For the rarefied datasets, we used the median value over the 100 iterations for each species. We rounded any non-integer species abundances, and removed any species that had no non-zero observations left at a site. After modelling, we adjusted for RtM effects by adding the expected RtM effect for each initial abundance class to the slope estimates and to the estimates of the random effects. As expected, the RtM effect was strongest for the highest initial abundance intervals (Supplementary Table 5). We excluded one dataset (Datasource_ID 1396) because the taxonomy was not consistent over time. One model (abundance interval 2 with classification based on year 1) failed to converge, possibly owing to an excess of zeros, so we ran a simpler model excluding one study (Datasource_ID 1560) and removed the random effect on location. In a small test, this same simplified model provided similar results to those of other models, with a difference in mean slope of 0.00096 (4% of the 95% CI) and 0.00032 (10% of the 95% CI) for the abundance intervals 4 and 5.

**Sensitivity analyses.** We performed sensitivity analyses on the biodiversity metrics to test for the effect of time-series length and the number of sites per dataset, as well as the influence of Europe and North America, from which most data originated.

To see the effect of time-series length on the mean slope estimates, we (1) selected only the sites with at least 20 years between the start and the finish of the time series; and (2) used only the last 10 years of each time series (discarding all sites for which fewer than 10 years between start and end remained). We ran our standard model (above) for the variables abundance, richness and Simpson diversity (see Extended Data Fig. 2).

For the effect of the number of sites per dataset, we retained only the 50, 20 and 10 best sampled (highest number of years data) sites per datasets. All other sites were discarded. We repeated our standard model (above) for the variables abundance, richness and Simpson diversity (see Extended Data Fig. 3).

To test the effect of the well-sampled continents Europe and North America, we ran our standard model for the variables abundance, richness and Simpson diversity for the data from Europe and North America, and for the data excluding these continents. The result of this analysis is presented in Extended Data Fig. 8.

**Conceptual model.** The aim of this simple model was to show possible changes in biodiversity metrics and population abundances under three scenarios: (1) all species decline proportionally to each other; (2) abundant species decline more than rare species; and (3) rare species decline more than abundant species. To do this, we created an idealized SAD of 43 species as starting values for each scenario. We chose abundances such that there would be realistic numbers of species in each of five starting SAD intervals (12 singletons, 10 doubletons and 21 species with higher abundances, with a highest abundance of 32).

In scenario 2, the annual decline of each species was proportional to its starting abundance ($N_t = N_{t=0}^{1-0.018t}$, where $N_0$ = the number of individuals at time 0, and $t$ is the time of measurement). Here, the most

abundant species declined by around 6% per year, and the rarest species did not decline. In scenario 3, the decline of the rarest species was around 6% per year and that of the most abundant species 0.76% per year: $N_t = N_{t=0} \times (10^{-0.027} + 0.035 \times \log_{10}(N_{t=0}))^t$.

We extrapolated these species abundance changes for 20 years, added a random error of ($\mu = 0, \sigma = 0.5$) to the yearly values, and rounded them to integers. The error was not intended to be a realistic simulation of population dynamics, but rather to add some variation for statistical model fitting. For visualization, we declared all species with an abundance below 0.6 to be extinct (abundance = 0).

We analysed these scenarios using the same approach as for the real data. All code is available at https://doi.org/10.5281/zenodo.8369189 (ref. 31).

## Reporting summary

Further information on research design is available in the Nature Portfolio Reporting Summary linked to this article.

## Data availability

All analytical code and data (Supplementary Data 1 and 2) used for the analysis are available at Zenodo: https://doi.org/10.5281/zenodo.10115304 (ref. 31), and the metadata for these files is available in Supplementary Methods 1. The raw data are available at https://doi.org/10.5063/F1ZC817H (ref. 22) in so far as they are not openly available elsewhere.

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

**Acknowledgements** We thank R. Didham for helpful comments; N. Naderi and A. Al-Hemiary for help with the digitization; M. Desquilbet and L. Gaume for thoroughly checking our dataset for instances of errors; A. Sagouis and T. Engel for help with the R code; A. Alzate for help with the Spanish translations; H. Schenk for help with the conceptual model; and S. Blowes for advice on the biodiversity metrics used. Funding: German Research Foundation grant FZT 118 (R.v.K., D.E.B. and J.M.C.) and China Scholarship Council (CSC) 202104910063 (M.S.). The following funding sources were used to collect the original data we used in our analysis: National Science Foundation NSF06-20443, 8811906, 9411976, 0080529, 0217774, DEB-0423704, DEB-1633026, DEB-1637685, DEB-1256696, DEB-0832652, DEB-0936498, DEB-1832016, DEB-0620652 DEB-1234162, OCE-9982133, OCE-0620959, OCE-1237140 and OCE-1832178, and the German Research Foundation DFG Priority Program 1374 'Infrastructure Biodiversity Exploratories'.

**Author contributions** Conceptualization: R.v.K., D.E.B. and J.M.C. Methodology: R.v.K., J.M.C., D.E.B., M.S. and S.R.S. Investigation: R.v.K., J.M.C., K.B.G., M.S. and S.R.S. Visualization: R.v.K. and D.E.B. Writing (original draft): R.v.K. Writing (review and editing): J.M.C., D.E.B., K.B.G., M.S. and S.R.S.

**Competing interests** The authors declare that they have no competing interests.

**Additional information**
**Correspondence and requests for materials** should be addressed to Roel van Klink.

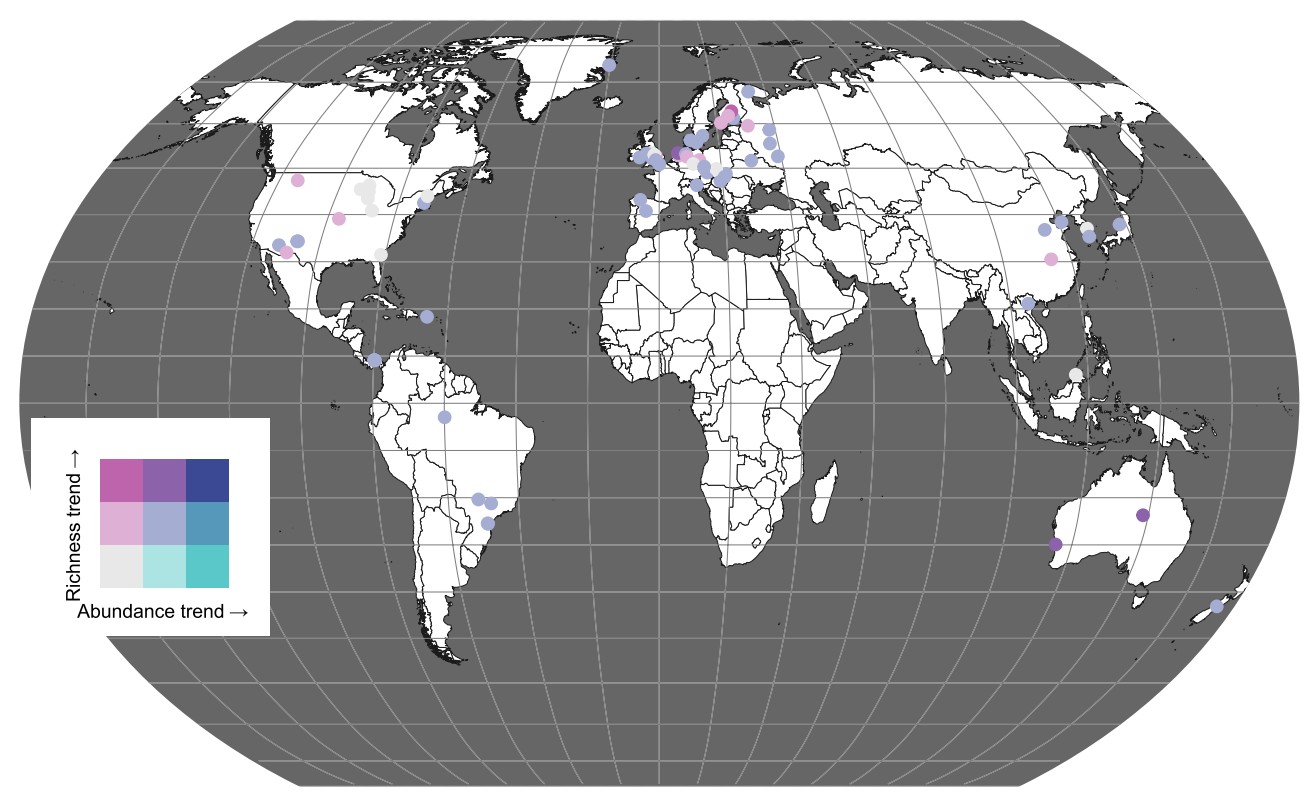

**Extended Data Fig. 1 | Bivariate map of changes in richness and abundance at the study level.** The trend estimates of the individual studies were derived from the random effects of the hierarchical Bayesian model. The colours are coded according to the strength of evidence, in which the middle colour in both axes indicates that the 80% CI includes zero, hence, all other colours indicate weak to strong evidence of a temporal trend on either abundance, richness or both. Map derived from refs. 152 and 153.

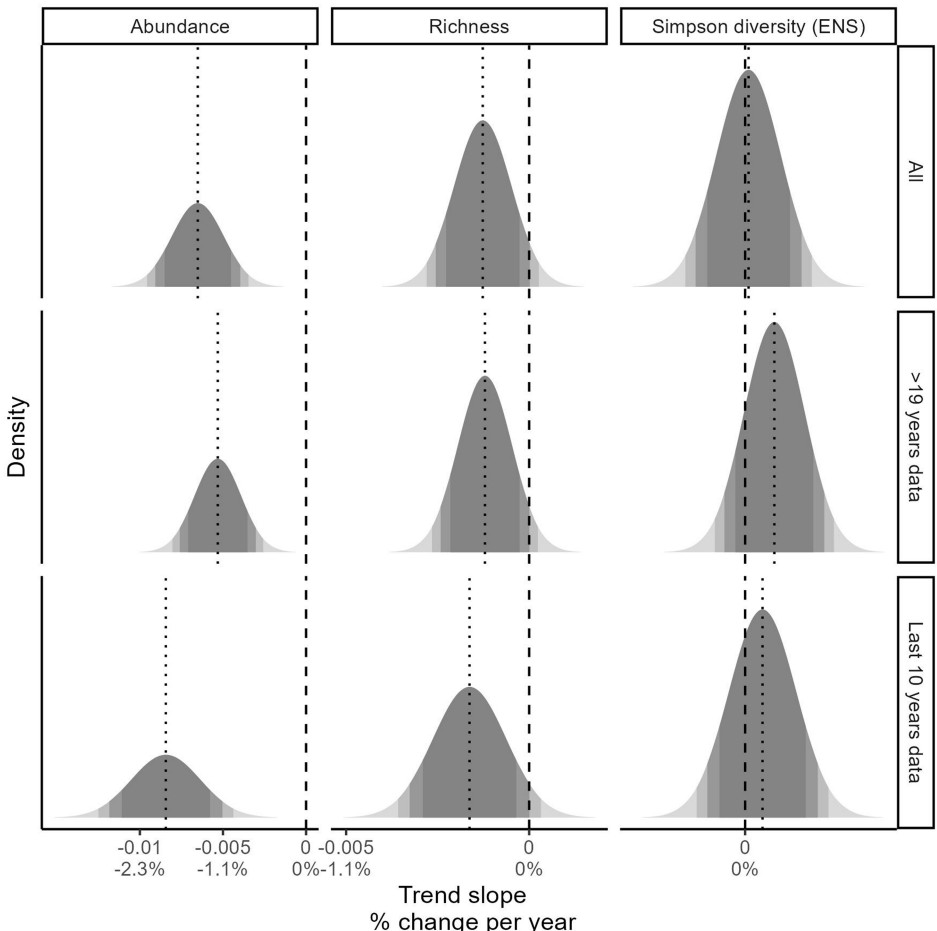

**Extended Data Fig. 2 | Influence of time-series length on the estimated temporal slopes of abundance, richness and Simpson diversity.** Long time series were selected by restricting the data to sites with at least 20 years from the first to the last year. Short time series were created by only retaining the last 10 years of each site. To aid the comparison among rows, the mean estimates of each realm are provided as dotted lines. This shows some shifts in mean estimates, but no differences in the qualitative interpretation of the results.

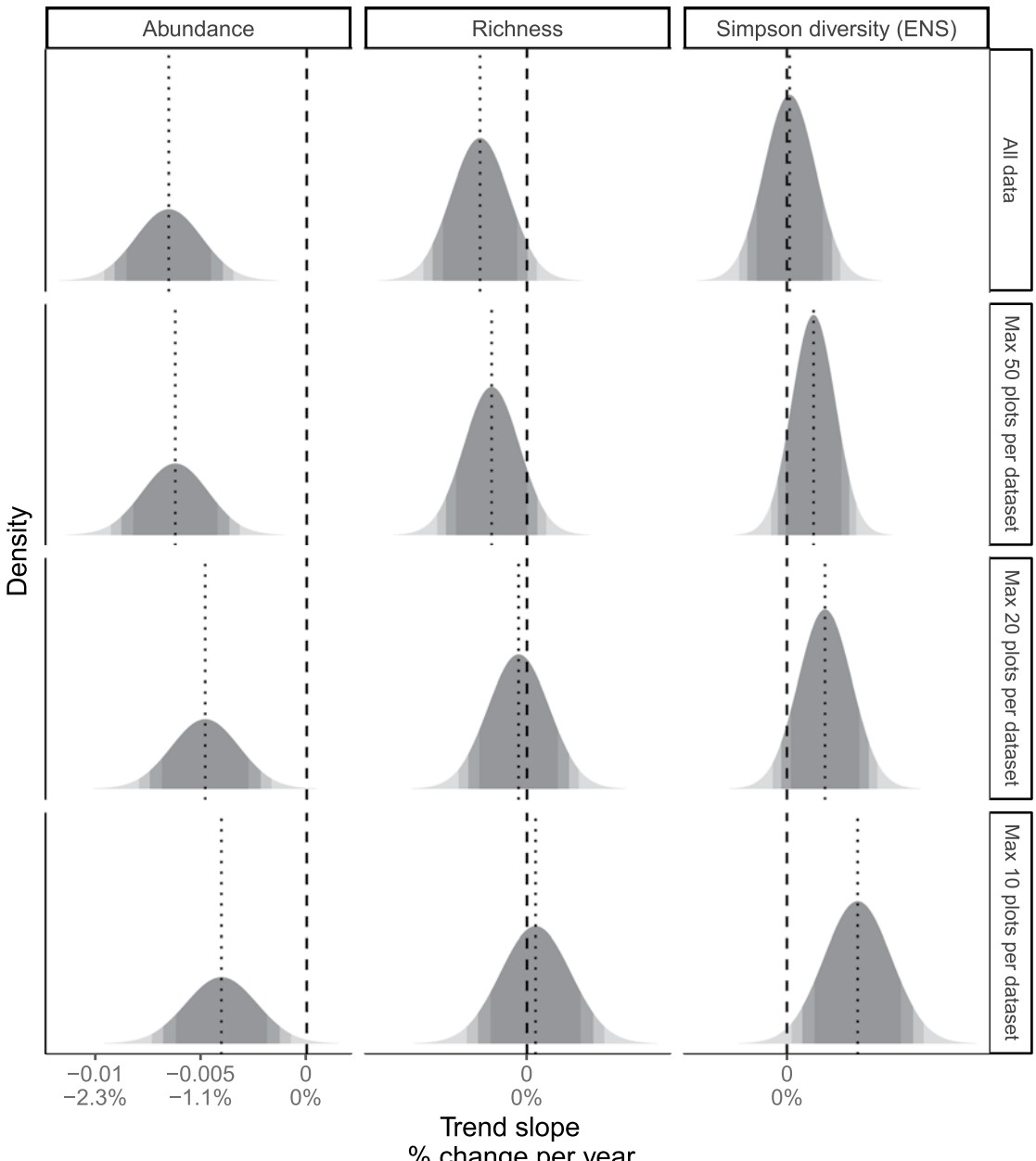

**Extended Data Fig. 3 | Influence of studies with a large number of sites on the estimated temporal slopes of abundance, richness and Simpson diversity.** The studies underlying the analysis were subsetted to only the 50, 20 and 10 best sampled sites, and the models were rerun. To aid comparison among rows, the mean estimates of each realm are provided as dotted lines. This shows a progressive shift to more positive slopes for all metrics.

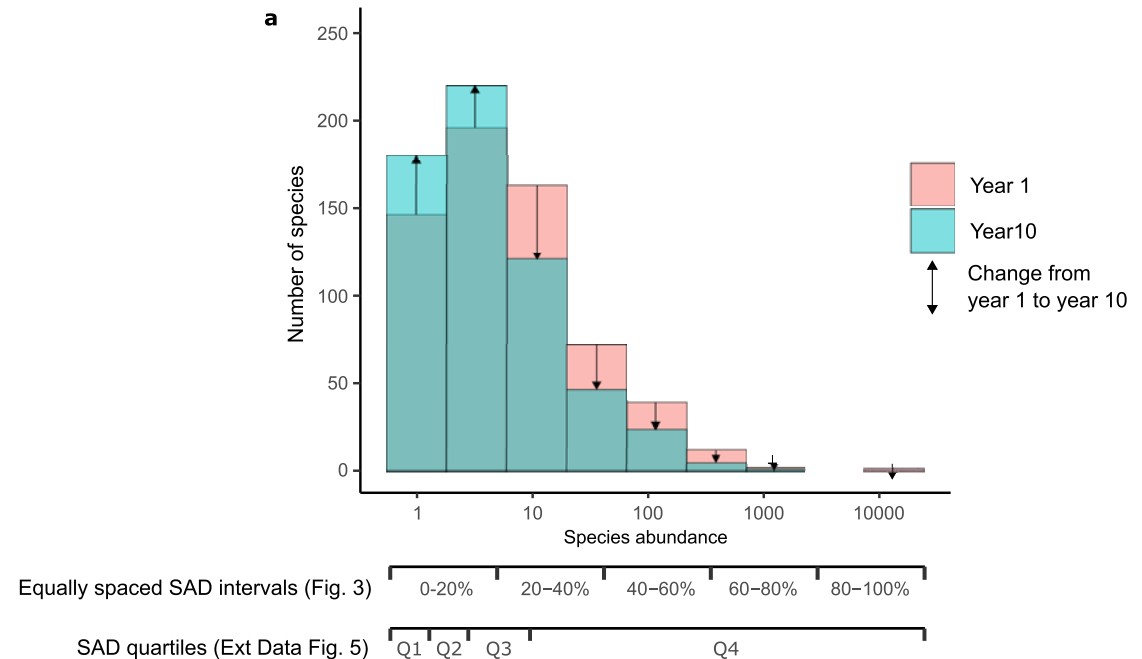

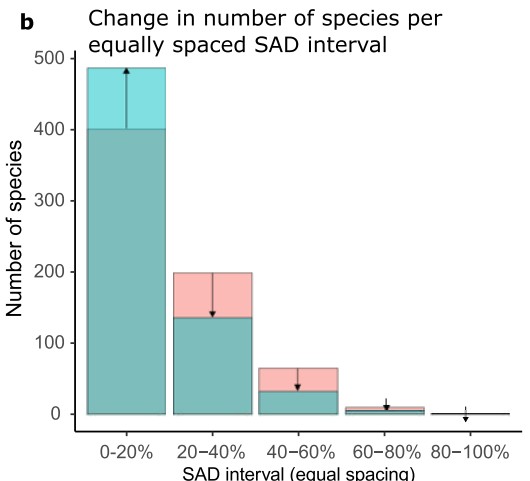

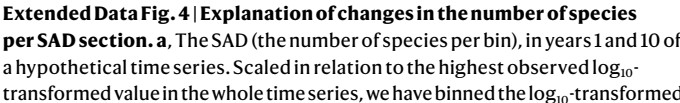

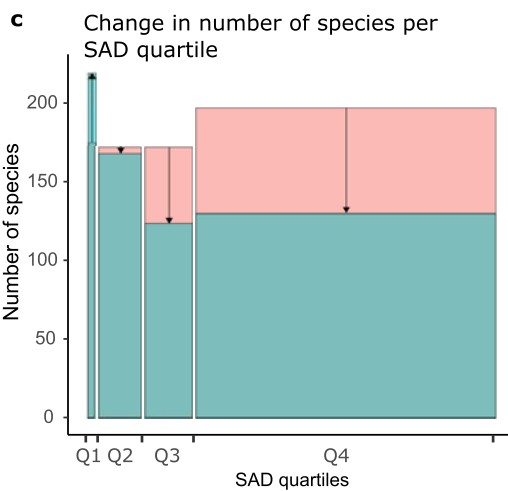

**Extended Data Fig. 4 | Explanation of changes in the number of species per SAD section. a**, The SAD (the number of species per bin), in years 1 and 10 of a hypothetical time series. Scaled in relation to the highest observed $\log_{10}$-transformed value in the whole time series, we have binned the $\log_{10}$-transformed species abundance values in two ways. **b,c**, We assigned five equally spaced sections (**b**), and four quartiles (**c**), where the quartiles had approximately equal species numbers (variation because of rounding of bins). After the bins were assigned, we calculated the number of species falling in each bin in each year.

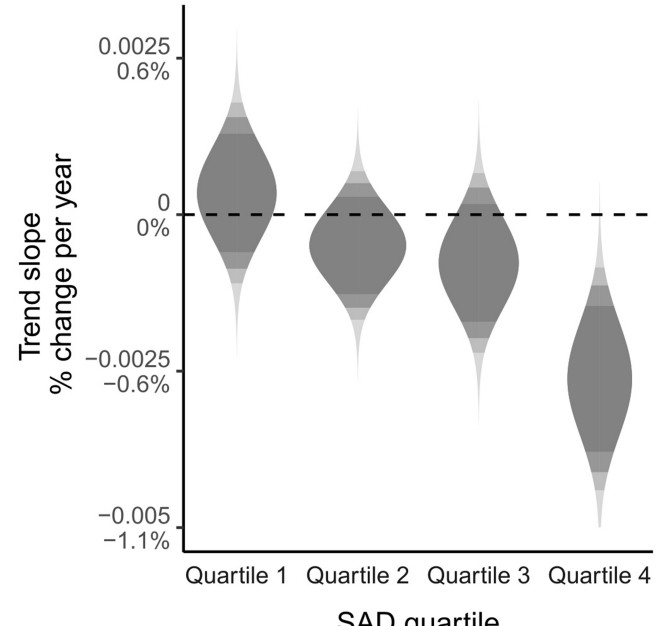

**Extended Data Fig. 5 | Changes in the number of species in each of four quartiles on the basis of the distribution of values of all species in each whole time series.** In comparison to Fig. 4, there are more species and more individuals in the higher quartiles, given the naturally low number of very abundant species (see Extended Data Fig. 4a,c). Here, there is an equal number of observations in each quartile, whereas in Fig. 4, the spacing (in log space) between bins is equal.

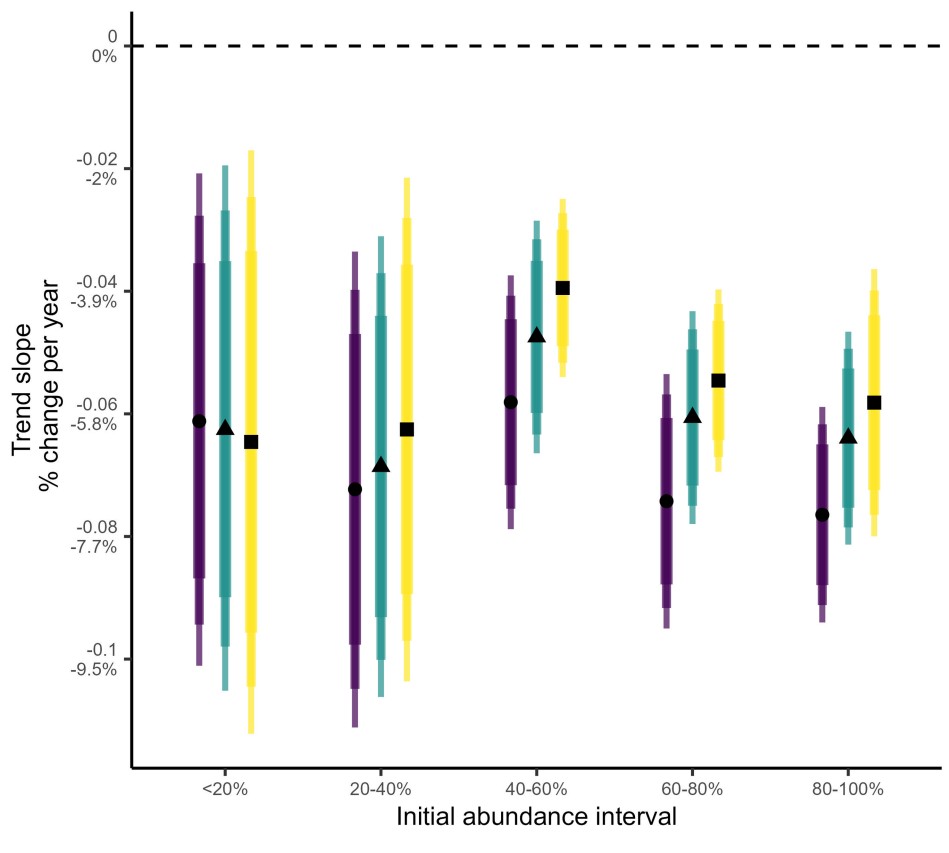

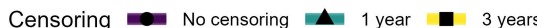

**Extended Data Fig. 6 | Effects of censoring the first year or first three years on the slope estimates for correcting RtM effects.** We calculated the slope estimate (±80, 90 and 95% CI) on the highest quality datasets (260 plots in 26 datasets with at least 15 years of data) for each of the initial abundance groups with no censoring (all data included), excluding the first year, and excluding the first three years. The shrinkage towards zero of the estimates with increasing censoring is assumed to be caused by RtM effects, but we cannot exclude that it's partially due to true greater declines of populations during the early years of monitoring. We have taken the difference between the mean slope estimate without censoring and the estimate with one-year censoring from the start as the correction factor for RtM effects in the main text (see Supplementary Table 5). Three-year censoring would provide a larger correction factor (Supplementary Table 5).

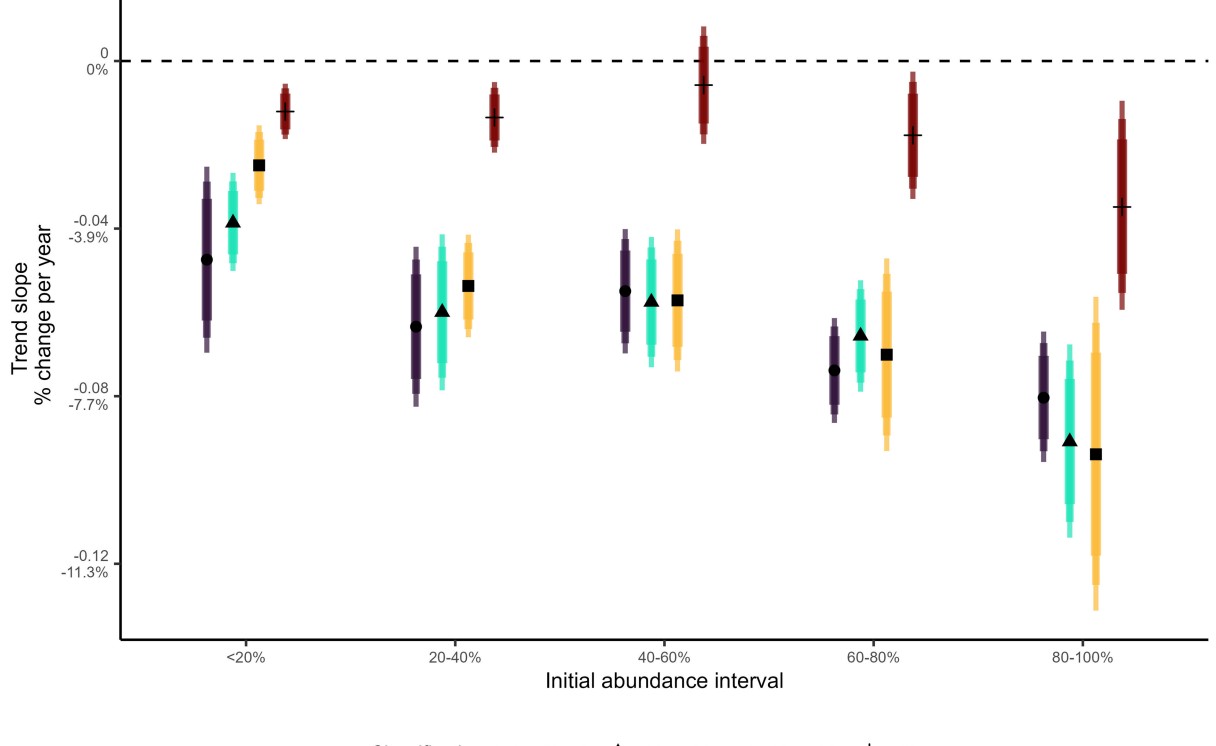

**Extended Data Fig. 7 | Effects of different ways of classifying locally rare and abundant species on the estimated mean population trends.** Classification of species starting interval was done based on (i) the abundances observed in year 1, (ii) the abundance of each species averaged over years 1 and 2 (iii) the abundance of each species averaged over years 1-5, and (iv) the abundance of each species averaged across the whole time series. Species that were absent at the start of the time series were not analysed, because their abundance trends will be, on average, positive by definition. The weakest results when based on the whole time series was expected because this is the most conservative approach to assessing rarity. An initially abundant species with a strong decline might not be abundant across the whole time series. Hence, this analysis shows that even when looking at abundance across the whole time series, the most abundant species decline most strongly.

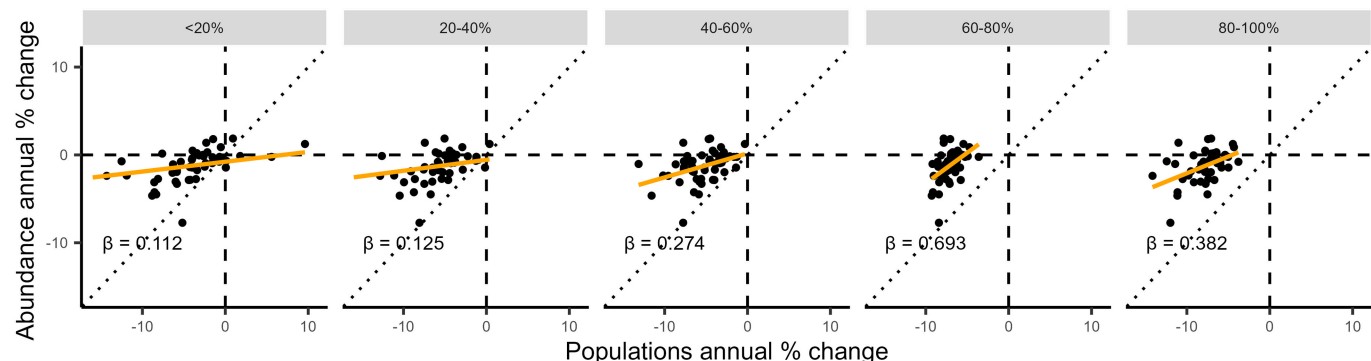

**Extended Data Fig. 8 | Relation between abundance slopes and the mean population slopes at the dataset level.** Abundance slopes and mean population slopes at the dataset level were both converted to the percentage change per year per initial abundance interval. The initial abundance intervals can be understood as the abundance interval of a species in relation to the log-transformed abundance of the most abundant species in year 1. Dotted lines represent the 1:1 relation, orange lines and slope estimates ($\beta$) represent the best fit according to least squares regression models. We removed one data point with an extreme slope from panels 1 and 2 to aid visual interpretation.

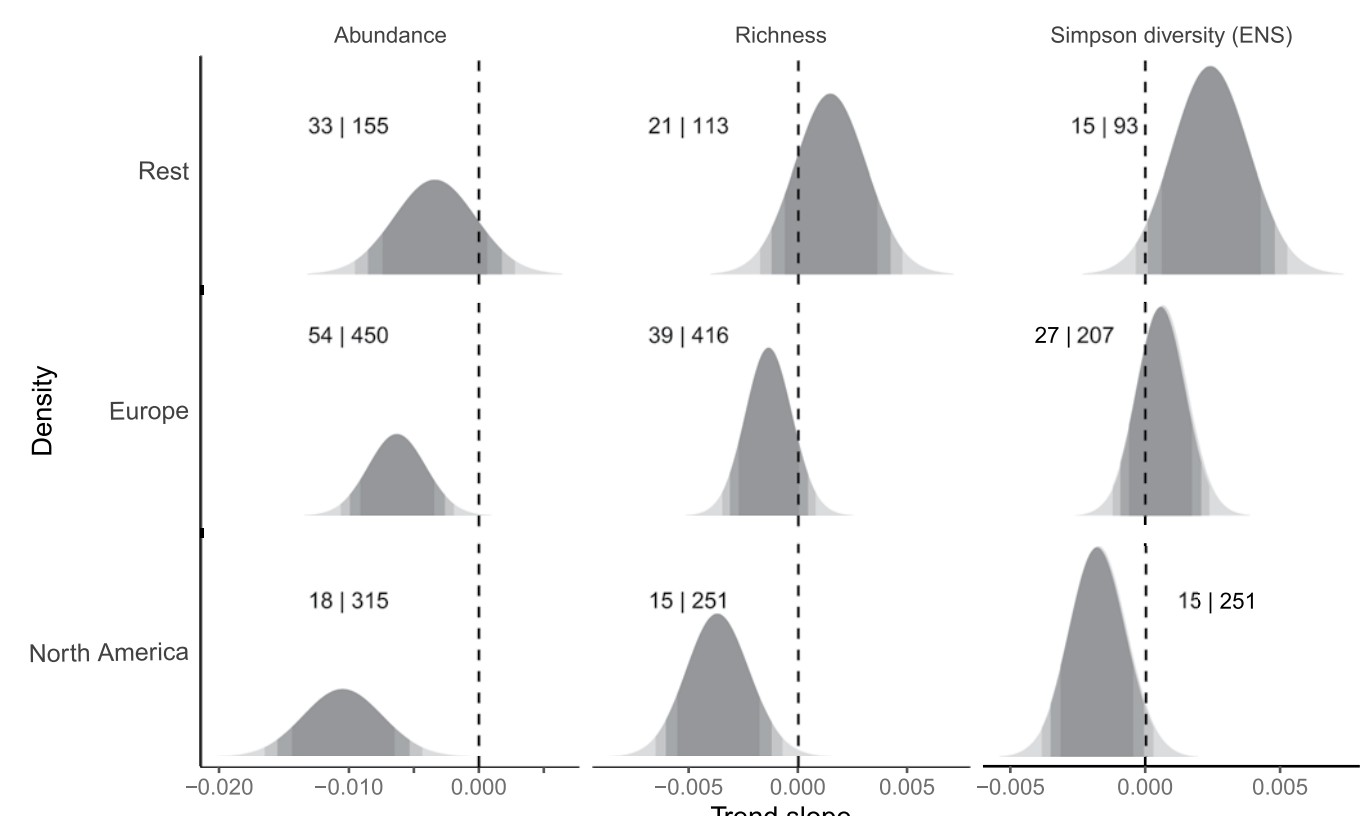

**Extended Data Fig. 9 | Temporal trend slopes for biodiversity metrics excluding Europe and North America.** The probability densities are shown for the slope estimates of abundance, species richness and Simpson diversity (ENS) for the data excluding Europe and North America. Numbers indicate the number of studies and the number of sites underlying each estimate respectively.

# Reporting Summary

## Statistics

For all statistical analyses, confirm that the following items are present in the figure legend, table legend, main text, or Methods section.

| n/a | Confirmed | |
|---|---|---|
| ☐ | ☒ | The exact sample size (*n*) for each experimental group/condition, given as a discrete number and unit of measurement |
| ☐ | ☒ | A statement on whether measurements were taken from distinct samples or whether the same sample was measured repeatedly |
| ☐ | ☒ | The statistical test(s) used AND whether they are one- or two-sided *Only common tests should be described solely by name; describe more complex techniques in the Methods section.* |
| ☐ | ☒ | A description of all covariates tested |
| ☐ | ☒ | A description of any assumptions or corrections, such as tests of normality and adjustment for multiple comparisons |
| ☐ | ☒ | A full description of the statistical parameters including central tendency (e.g. means) or other basic estimates (e.g. regression coefficient) AND variation (e.g. standard deviation) or associated estimates of uncertainty (e.g. confidence intervals) |
| ☐ | ☒ | For null hypothesis testing, the test statistic (e.g. *F*, *t*, *r*) with confidence intervals, effect sizes, degrees of freedom and *P* value noted *Give P values as exact values whenever suitable.* |
| ☐ | ☒ | For Bayesian analysis, information on the choice of priors and Markov chain Monte Carlo settings |
| ☐ | ☒ | For hierarchical and complex designs, identification of the appropriate level for tests and full reporting of outcomes |
| ☐ | ☒ | Estimates of effect sizes (e.g. Cohen's *d*, Pearson's *r*), indicating how they were calculated |

*Our web collection on statistics for biologists contains articles on many of the points above.*

## Software and code

Policy information about availability of computer code

| Data collection | Some data were digitized using ImageJ software, and some were digitized using the MetaDigitise package for R. All data were processed in R between 2018 and 2023 |
|---|---|
| Data analysis | We used R 4.2.2 for all data processing , and the INLA package for statistical analyses. Visualization was done using the  GGplot2 package. We used hierarchical Bayesian modeling to estimate the temporal slope of the various biodiversity metrics (i.e. the explanatory variable w as 'Year' |

For manuscripts utilizing custom algorithms or software that are central to the research but not yet described in published literature, software must be made available to editors and reviewers. We strongly encourage code deposition in a community repository (e.g. GitHub). See the Nature Portfolio guidelines for submitting code & software for further information.

## Data

Policy information about availability of data

All manuscripts must include a data availability statement. This statement should provide the following information, where applicable:
- Accession codes, unique identifiers, or web links for publicly available datasets
- A description of any restrictions on data availability
- For clinical datasets or third party data, please ensure that the statement adheres to our policy

All Data as analyses in this paper are available as supplementary material, and the underlying raw data are available on Knowledge Network for Biocomplexity. The

# Research involving human participants, their data, or biological material

Policy information about studies with human participants or human data. See also policy information about sex, gender (identity/presentation), and sexual orientation and race, ethnicity and racism.

| | |
|---|---|
| Reporting on sex and gender | NA |
| Reporting on race, ethnicity, or other socially relevant groupings | NA |
| Population characteristics | NA |
| Recruitment | NA |
| Ethics oversight | NA |

Note that full information on the approval of the study protocol must also be provided in the manuscript.

# Field-specific reporting

Please select the one below that is the best fit for your research. If you are not sure, read the appropriate sections before making your selection.

☐ Life sciences          ☐ Behavioural & social sciences          ☒ Ecological, evolutionary & environmental sciences

For a reference copy of the document with all sections, see nature.com/documents/nr-reporting-summary-flat.pdf

# Ecological, evolutionary & environmental sciences study design

All studies must disclose on these points even when the disclosure is negative.

| | |
|---|---|
| Study description | We evaluated the biodiversity change of assemblages of terrestrial insects, arachnids and Entognatha (springtails and allies) over time to understand the patterns underlying insect declines. We performed this study by means of synthesis of existing and openly accessible data. |
| Research sample | Our aim was to evaluate as many datasets as were available under the following restrictions: We collected time series of insect/arachnid/Entognatha assemblages (at least all individuals of a family or higher taxonomic level, counted), assessed using consistent methodology  with at least 9 years between the first and last sampling year, that were or could be made openly accessible. A low proportion of non-target taxa - other arthropods such as crustaceans and myriapods(<10%) was acceptable, whenever these could not be separated from the target taxa.) Our sample size was thus dependent on the amount of openly available. We used 106 datasets. The exact provenance of each study is detailed in Extended Data table 5.<br>The sample population is the world, but we are aware that our data are biased in various ways. Therefore, the data and analysis mostly represent those locations from which data are available. We show the biases in several sensitivity analyses |
| Sampling strategy | The sample size was determined by the availability of data. When necessary and possible, we used rarefaction methods to equalize sampling effort across years. Experimental treatments (where researchers actively manipulated environmental conditions) were excluded. |
| Data collection | Standardized and non-standardized data searches were performed by Roel van Klink, Jonathan Chase and Konstantin Gongalsky. Data were digitzied, extracted and standardized by Roel van Klink, Minghua Shen, Abdul Al-Hemiary and Nina Naderi. The collection of the original data involved numerous data collectors and data analysts. |
| Timing and spatial scale | Data searches were performed from 2018 - 2021, but the collated data go back to 1951,Data from around the world were included, but there is a bias towards Europe and North America (detailed in the paper) |
| Data exclusions | No data exclusions other than data or years that did not fit our analysis criteria (mostly in cases of taxonomic or methodological inconsistencies). |
| Reproducibility | All code and analytical data for statistical tests is available on Zenodo https://doi.org/10.5281/zenodo.8369189. All raw data can be found at KNB: https://knb.ecoinformatics.org/view/urn%3Auuid%3Ab338c276-1d3f-4cc7-a192-cad846083455 |
| Randomization | NA (no treatments were analysed) |
| Blinding | NA |

Did the study involve field work?          ☐ Yes          ☒ No

# Reporting for specific materials, systems and methods

We require information from authors about some types of materials, experimental systems and methods used in many studies. Here, indicate whether each material, system or method listed is relevant to your study. If you are not sure if a list item applies to your research, read the appropriate section before selecting a response.

## Materials & experimental systems

| n/a | Involved in the study |
|-----|------------------------|
| ☒ ☐ | Antibodies |
| ☒ ☐ | Eukaryotic cell lines |
| ☒ ☐ | Palaeontology and archaeology |
| ☒ ☐ | Animals and other organisms |
| ☒ ☐ | Clinical data |
| ☒ ☐ | Dual use research of concern |
| ☒ ☐ | Plants |

## Methods

| n/a | Involved in the study |
|-----|------------------------|
| ☒ ☐ | ChIP-seq |
| ☒ ☐ | Flow cytometry |
| ☒ ☐ | MRI-based neuroimaging |

