## [Peer Review File · Nature]

Manuscript Title: Disproportionate declines of formerly abundant species underlie insect loss

Reviewer Comments & Author Rebuttals

Reviewer Reports on the Initial Version:

Referees' comments:

Referee #1 (Remarks to the Author):

Review of MS 2022-06-08651 Nature

Widespread declines of dominant insect species change assemblage's structure

van Klink et al.

4th July 2022

The manuscript presents a global synthesis of 171 long-term (>10yr) time series of insect abundance change in terrestrial and freshwater systems. The 'global insect decline' theme is extremely topical and important, and this manuscript represents an in-depth analysis of an exceptional dataset.

I must admit, though, that I really struggled to see the major new advance that the authors were trying to put forward.

Yes, the database has been expanded from the 148 studies in the authors' previous syntheses (Refs 13,18) to 171 studies here, through the use of additional search parameters (and publication languages). This is important, but not a major advance.

Yes, it is important that there were contrasting biodiversity trends identified between freshwater and terrestrial systems, but this was a key conclusion of the authors' earlier work (Ref 13).

Yes, it is important to try and break down these assemblage-level responses into contrasting responses across common vs rare species, and I like the general approach of looking at whether different biodiversity metrics (richness, evenness, dominance etc) are 'decoupled' due to idiosyncratic species responses in different systems. However, the concept that metrics can be 'decoupled' in the face of perturbation is not new (eg Ref 8), and some of the authors are involved with the more extensive study of this in the biorxiv pre-print by Blowes et al. (Ref 7).

Actually, I completely agree with the one of the Blowes et al conclusions that it is important to look beyond overall measures of 'abundance change' or 'richness change' because these can mask important (potentially contrasting) underlying processes. This is certainly what the current manuscript is trying to achieve, but the final outcome felt much more like an affirmation of concepts and trends that were already known, rather than anything that was really unexpected (to me).

The question is whether the headline result, of steeper trend slopes of decline in the number of species in the dominant abundance-class is sufficiently noteworthy. Refs 23 onwards give many specific examples of common/dominant species declining in abundance through time, but perhaps the generalization of this result for a compilation of 171 studies carries more weight? I am of two minds about this. It is a nice pattern (ie Fig 2, notably), but it is pretty much just presented as that; a nice pattern in the data, with no deeper insight about the processes or drivers. So, in a sense, the cited studies that give specific findings about decline/increase in individual study systems for individual species, are more informative in many ways. For a synthesis like this I would have expected some broader insight that could not be gained from the component studies, and I felt that was lacking here.

In addition, I had quite a number of problems with the ways in which the analyses were conducted, and as a consequence I did not find the general conclusions overly compelling, or robustly proven.

Stepping through the analytical issues, using the Figures as focal points:

Fig 1 looks lovely at face value, but is disappointingly uninformative on deeper inspection. As the authors note themselves Fig 1b just reflects a trivial sampling process in which richness increases with sample abundance. This phenomenon is so well known and has little bearing on biological interpretations of the data, so I can't see any justification why Fig 1b should be in the main text, rather than in the extended data figures or supplement.

The main panel to Fig 1a presents abundance and richness trends for freshwater and terrestrial studies in the upper two frequency distributions, which is fine. The numbers to the right of the plots seem to represent number of studies/sites for each metric, and that raised an immediate red flag about the comparison of metrics based on differing numbers/identities of studies compiled. Across metrics, Ext Dat Table S1 and Fig 1a seem to suggest that there might only be 54 terrestrial studies and 20 freshwater studies (or fewer) out of 171 that would have all four comparative metrics of abundance, richness, rarefied richness and evenness for the same suite of studies (?). Given the earlier preamble in the Introduction about 'decoupling' of metrics, it would seem important to identify the subset of studies in which it is actually possible to tease apart *all* the metrics, and perhaps present that as a sensitivity test in the supplement.

For the remainder of Fig 1a (the rarefied richness and evenness frequency distributions), I don't think the metrics used are really fit for purpose.

In the Methods section ('Univariate metrics') the authors indicate that rarefied richness was interpolated at the lowest total abundance observed in any year in the timeseries (except when lower than 10 individuals, in which case rarefied richness was extrapolated to a 10 individual sample). Rarefying in this way throws away most of the information in the data, and forces a comparison among studies at potentially very low sample abundances, right near the steeply-ascending basal portion of the species accumulation curve. We don't know how serious a problem this is, because we are not given the information about the abundance level at which the rarefaction values were calculated for each study. In most cases, though, there will be almost no power to discriminate richness trends using this rarefaction approach (hence, no surprise that there appears to be little resolution in rarefied richness trends).

Later in the methods (under 'Preparation of raw data'), the authors use a different approach to standardisation of sampling across years, which is to subsample in order to equalise sampling effort (eg number of trap samples), rather than equalising sample abundance (as in the case of rarefaction above). They go on to describe some examples of situations in which the data were wrangled into shape based on standardising sample effort, but this very much sounds like a subjective set of approaches that will be inconsistent across studies and analyses.

I would suggest changing to a more objective approach for all these types of standardisation (rather than by sample abundance or by sampling effort), and instead standardise objectively by sample coverage (eg see papers by Anne Chao, and the recent Roswell et al 2021 paper in *Oikos*). Sample coverage intrinsically recognises that small sample abundances and/or low sample sizes could actually accumulate a better 'coverage' of a small total fauna than large samples taken from a much larger total fauna (so, it's not only about the size of the sample, but about the degree to which it captures the species pool in a representative manner). There are R packages like 'iNEXT' or 'meanrarity' that could interpolate/extrapolate to a common sample coverage value (like 0.8 or whatever), which would be more objective and use more of the available data (in a more consistent manner).

Next, for evenness in Fig 1a, I have even bigger concerns. The authors describe their evenness measure as ENS-PIE "Hurlbert's probability of interspecific encounter (= inverted Simpson index), converted to the effective number of species". The authors also state in the methods "We calculated a number of biodiversity change metrics based on the framework of Hill-numbers (Ref

21). This framework unifies different metrics of biodiversity, evenness and compositional change". This set of statements does not make sense to me, as written.

First, Hurlbert's PIE is related to the *complement* of Simpson's index (ie $1 - \sum(p^2)$), not to inverse Simpson ($1/\sum(p^2)$). Moreover, the complement of Simpson's index is otherwise known as the Gini-Simpson diversity index, and is not (in its own right) a measure of evenness. It is a measure of diversity. There is also an important distinction that Hurlbert's PIE is calculated by sampling *without replacement* whereas the complement of Simpson's is calculated by sampling *with replacement* (and this makes a big difference when the sampling universe is finite, such as in many of the small-sample cases that the authors describe in their data). Then, the conversion to 'effective number of species' (ENS) is presumably what the authors imply by their use of the Hill series framework [ie inverse Simpson, $1/\sum(p^2)$, is indeed the effective number of species for the Simpson measure]. Of course, this ENS measure is also based on sampling with replacement, and crucially is a diversity measure too, not an evenness measure. Of all the Hill series 'numbers equivalents' of diversity measures that the authors could have chosen, it is also the one that is most highly influenced by the dominant species in the assemblage.

I think this is a major problem. If the authors claim to be using the Hill series framework, then a quick read of Hill 1973 would show that Hill proposed evenness measures that are a ratio of any two ENS 'numbers equivalents', such as $E_{1,0}$, or $E_{2,1}$ (the latter for example being the ratio of inverse Simpson divided by exponential Shannon). Classics like Smith & Wilson 1996 A Consumer's Guide to Evenness Indices, Oikos, provide useful information on this, and alternative evenness measures (they advocate Evar, for instance). There are many different options for a genuine evenness measure that could be adopted, rather than the 'diversity' measure that the authors use here, masquerading as evenness.

Of course, the Hill series 'numbers equivalents' of diversity are useful to use in their own right (as diversity indices), but I would suggest not just using inverse Simpson (the most affected by dominant species), but rather the range of metrics right across the Hill series that provide differential weight to rare vs common species. Read the Roswell paper for additional insight on this.

Figure 2 – trend slopes by abundance percentile class in the species-abundance distribution: This is the only really important figure in the manuscript, and the one that seems to have the fewest analytical problems associated with it.

The authors have done their due diligence in appropriate sensitivity tests of different ways of slicing up the SAD distribution (either 20% abundance intervals, as shown in Fig 2, or equal numbers of species per quartile in Fig. S6), and the kernel density estimate to show changes in the shape of the SAD in Fig 2b is effective.

I do wonder about a couple of things stemming from 'relativising' commonness vs rarity at the plot level. One is that the same study could have multiple plot/site locations, and relativising a species as 'common' (ie 80-100% abundance interval) in plot 1, where all species have low abundances because the site is disturbed, seems quite different than the same species being designated as 'common' in an adjacent natural site Plot 2 where total assemblage size is much larger. Absolute abundances would be very different between Plots 1 and 2 for this species, but in both cases the SAD relativisation would potentially deem them equally common.

The authors will no doubt argue that this doesn't matter because it is about the relative trend through time at each plot (separately). However, it does matter because the authors key conclusion is that the *number of species* in the 'common' 80-100% SAD interval is declining through time, but that is definitely not the same thing as designating an overall classification of 'common' vs 'rare' and then investigating the *abundance* trends of the common species to show that their rate of decline in abundance is steeper than for species in rarer abundance classes. In the original 171 studies, authors would have used the absolute abundance data from multiple plots (eg disturbed vs undisturbed sites etc) to infer abundance trends through time for each species, and my gut feeling is that the authors approach in this manuscript showing richness changes per SAD abundance class would not mirror abundance trends through time estimated at the species level.

Lastly on this point, I would also say that most readers would be misled by the title of the

manuscript, "Widespread declines of dominant insect species", which implies that the authors are reporting abundance declines of common species, which is a different thing than what they show in Fig 2 (declining richness of common species at a site).

Figure 3 – species turnover

I was quite surprised that such a crude approach was used for species turnover through time. At one level it is good to contrast the qualitative presence-absence metric with a quantitative abundance metric, but the Morisita-Horn metric is notable for being strongly influenced by the most abundant species in the assemblage. In this sense, yes it is a match to the inverse-Simpson, but that is not a good thing in terms of a conservative approach to assessing whether changes in dominant species are driving assemblage-wide restructuring. The use of Morisita-Horn represents an undue bias toward the intended outcome. Morisita-Horn also does not handle joint absence similarities as well as some other dissimilarity metrics either.

As a second major problem, the turnover values are calculated as repeated pairwise dissimilarities to the single year at the start of each time series. It should be self-evident from the recent insect decline literature that this will create a potential bias akin to a 'baseline effect'. Consider two examples: first, if the baseline year had an aberrantly dissimilar assemblage (visualised in an ordination plot this would be a point very far from the overall plot centroid) then all subsequent pairwise dissimilarities would be high, making it impossible to discern any trend in assemblage composition across the remaining years (whether one existed or not); second, imagine baseline assemblage composition close to the plot centroid value in an ordination, there could easily be a false appearance of gradually increasing pairwise dissimilarities through time, even if these were distributed in random vector directions each year (eg alternately increasing/decreasing along axes 1 and 2 of an ordination in different years, but not tending directionally along one ordination vector).

There is no way of telling what the magnitude of this problem is, but I would have no confidence whatsoever in the trends represented in Figure 3. There are many ordination methods for testing predictor effects on compositional trends in multivariate space. Exploring one or more of these approaches, along with a judicious left-censoring of time series to explore potential baseline effects would be well advised.

Referee #2 (Remarks to the Author):

The past decades, the biodiversity crisis has been frequently alerted to. Importantly, large scale data analyses have provided deeper insights into the rate and processes of biodiversity changes on our planet. The sometimes dramatic declines of insect abundance and its consequences for ecosystems functioning have been strongly quantified in more recent years. Metrics that have typically been presented to quantify biodiversity loss of insects have been total biomass, abundance, taxonomic richness as well as species turnover. Several studies have provided insights into how these metrics differ for realms, geographic areas or specific species groups. In their current study, van Klink and co-authors explore whether aquatic and terrestrial insect communities differ in biodiversity changes and how the rarity of insect species in communities is characterising the trends of biodiversity changes in the two realms differently. The study reveals that declines in dominant species in both realms occur, but that in aquatic systems (and not terrestrial) these are compensated by a rise in numbers of rarer species. These results are obtained by building an extensive database of reported studies that present longer term changes in biodiversity of their study sites. Overall the study presents a rigid data analysis, although some methodology could be presented in more detail (see specific comments below). The graphical presentation of the data transfers a clear message and is easily interpretable. I appreciate the use of alternative bins and approaches to show that the results are not sensitive to more arbitrary decisions in data analysis. My overall assessment of the study is that the study connects to perhaps the most important crisis we are currently facing, that of biodiversity decline and its consequences to ecosystems functioning, and the cascade in numerous societal consequences. It presents clear conclusions with rigid data analysis. At the same time, I have reservations in my enthusiasm for this study. These

include, i) novelty, in terms of extending our insights compared to published work, ii) overreaching of conclusions by the terminology/wording used, iii) unclarity of some sections in the main text and methods that make the manuscript less accessible to a broader readership (at least including myself).

i) The authors heavily built on their wonderful recent study in *Science* (2020; 368, 417-420), where they analyse a similar dataset for biodiversity metrics. Their previous work revealed that patterns in insect decline in terms of abundance differ for aquatic and terrestrial ecosystems. The same message is prominently part of the current manuscript, but deepened with analysis on metrics such as richness, evenness and species turnover. The responses of these metrics for aquatic and terrestrial realms show a pattern that is similar to work by Blowes et al (2019; *Science* 366, 339-345). Species turnover was found to be a dominant factor in biodiversity trends and that differences in turnover contributed to pattern differences for aquatic and terrestrial systems. Pilotto et al. (*Nature Communications*; 2020, 11:3486) similarly identified species richness trends that differ for terrestrial and aquatic systems and that taxonomic groups differ in trends of biodiversity metrics. Also this study shows the difference in responses of terrestrial and aquatic insect/invertebrate communities. In a more theoretical paper by Blowes et al. (*Ecology*, 2022, in press) the interactions between metrics are evaluated for their characterisation of biodiversity trends. The authors do acknowledge most of these and other studies in their manuscript. Although this new manuscript presents new analysis of common versus rare species, it also tells much of what other studies have shown. I do certainly appreciate the overall work, but also had expected a more significant novel message when placing this in context of *Nature*.

ii) Some of the statements, conclusion and title oversell the findings of the manuscript. By line number suggestions, I highlight some of these sentences below. Some of the strong statements are due to the choice of words. The term community structure is used by many ecologists to particularly describe species interactions, food web relationships and how assemblies form in specific areas, on shared resources etc. What the data is presenting are metrics of species observations, not about structure in terms of species relationships/interactions. When reading the title, I was expecting to see data on how loss of dominant species alters the interaction structure of an assembly. In the abstract the statement " abundance changes would lead to predictable changes in species numbers and community structure" , speaks differently to people working on keystone species, density mediated processes, priority/legacy effects in communities, as well as the role of biodiversity to stabilise communities (through structure). In these research fields it is a given that a change in abundance of one species may have very unpredictable consequences for community structure. I suggest to use the term biodiversity metric throughout the manuscript.

iii) In the introduction of methods in the main text as well as the methods section, some statements are technical and difficult to judge what it means for the analyses. For example, a) what does it mean for the reliability of your analyses when there is strong evidence with a 95% confidential interval, but weak evidence with 90 or 80%. Here you could help the reader by explaining what part of the story changes when using a different confidence interval. b) it is unclear to me how you can characterise abundance trends over time when you use a variable starting point in time, based on the highest abundance in the full time series. The explanation in the main text (Page 4, final paragraph) is too technical for me to understand how species are treated that have initial decline and recovery in later time points. c) in the methods I could not find how the authors calculated the rate by year, which is the central value used in all metrics. In the methods section, it would have helped me to make the rate by year very visible (perhaps even in a header).

By line number

Line 24: This statement is too strong. How abundance trends reflect changes in other biodiversity aspects is presented in many other papers that you cite, and thus not unknown.

Line 25: Here " community structure" is not the best term to use as at least part of the readers of

your paper would disagree with the statement.

Line 38: there are also many ecological studies that show that dominance of a species is not a good predictor of how it affects community structure / ecosystem function.

Line 55: here the term community structure should be avoided

Line 75-77: Here I suggest to make the more technical statement into a statement of what it means for the confidence of your ecological statements.

Line 82: This sentence seems incomplete

Line 83: The second part of the same sentence is not clear on what is meant with " lower mean estimates" of which parameter? Larger studies could be defined for number of sites or time series.

Line 131: here you could use rare / abundant species also in statements such as " strong evidence for declines of the upper 10%"

Line 138: Here the richness trend of terrestrial species is considered constant. This however depends on the confidence interval used. An overall low yearly rate may still be a significant decline over a long time period.

Line 142-151: This section is written very technical and it is not easy for the reader to identify how your method provides an accurate view on the yearly rate of change in a community metric.

Line 170: remove first full stop before reference to extended data

Line 187: structure should be replaced by biodiversity metrics

Line 187-205: The discussion here connects well to literature and also hints that other studies have shown that especially the abundant species are in decline. This makes me conclude more strongly that the insights presented are less novel. This certainly does not make your study less valuable, but could indicate that the finding is not of the level of novelty searched by the journal.

Referee #3 (Remarks to the Author):

This paper is a successor to a highly influential recent paper in *Science* (2020 368:417-420), which presented a meta-analysis of trends in insect abundance, biomass and species richness in terrestrial and freshwater communities. This new manuscript extends the dataset (adding 23 new studies), revises one of the earlier findings (with declines in terrestrial species richness no longer being well supported), and explores additional aspects of biodiversity: evenness, abundance structure and temporal turnover. The manuscript is reasonably well written (especially the introductory section, which lays out the issues skillfully), and it reports some striking new results: an increased evenness in species' abundances over time in both terrestrial and freshwater arthropod communities, which the authors link to a decline in the number of very abundant species, while the number of rare species holds steady or increases. If those findings prove to be robust, the paper could make a substantial impact, contributing to the ongoing policy and public debate on this issue.

However, in its currently form I would have some qualms about recommending this manuscript for publication (in *Nature* or elsewhere).

(a) To begin with a relatively minor matter: the order of the reporting of main results (lines 78-11)

seems odd. The natural order is to go from (i) summed abundance to (ii) species richness to (iii) rarified richness, to (iv) evenness – and indeed that is the order used in the Figures (Figs 1, S3, S4, S5). The manuscript moves the species richness result first, and (understandably) gives it disproportionate emphasis, presumably because it differs from the result published earlier (with terrestrial richness declines no longer deemed significant). However, doing so disrupts the flow of the argument in my opinion.

(b) More fundamentally, the index used for measuring evenness (Hurlbert's PIE index, which is equivalent to the finite sample form of Simpson's 1-D) is sensitive to both richness and evenness. It's a fine index, but it is not properly speaking an evenness index (although others have sometimes called it one, e.g. Wall et al. 2018, *Biodiv & Conserv* 27: 395-415), but rather a measure intermediate between a pure richness index and a pure evenness measure. As species richness is already incorporated into the analysis (both in raw and rarified forms), it would be preferable to use an evenness index that was independent of richness. The authors express a desire to use an index from the Hill family of indices, and it's true that a variant of PIE (Simpson's 1/D, which I believe is equivalent to the "expected number of species" variant used here), is a member of that family of indices with an exponent of 2 (and thus more evenness-weighted than exp(Shannon) with exponent 1), but there is a proper evenness index in the Hill family: the Berger-Parker index, with exponent of infinity. There are far better evenness indices than Berger-Parker (see e.g. Smith & Wilson 1996, *Oikos* 76:70-82; Beisel et al. 2003, *Internat. Rev. Hydrobiol.* 88: 3-15; Tuomisto 2012, *Oikos* 121: 1203-1218; and many others). There are even evenness numbers derived from Simpson's index: Hill himself published one (Hill 1997, *Oikos* 79: 413-416), and a more widely used variant was proposed by Krebs (1989: *Ecological Methodology*): $(1/D)/S$. PIE might be included as a general-purpose diversity index (and one that is relatively insensitive to sample size), but if the goal is to reflect evenness per se, the index is a poor choice.

(c) The section on abundance structure is also in need of refinement, and of better explanation. The authors devote most of their attention (lines 119-131) to the number of species in equal-sized abundance categories, and find a decline in the number of the commonest species, and an increase in the number of rare ones. But they present this in misleading terms: phrases like "Widespread declines of dominant species" (title), "declines of (formerly) dominant species" (abstract), "the most abundant species disproportionately declining" and the "rare species disproportionately increasing" (lines 119-120) sounds like they indicate shifts in the abundance of these species, rather than in the numbers of species in each category. These categories are themselves not fixed, but rather scaled relative to the commonest species in each dataset. Meanwhile, the Species-Abundance distributions derived from the kernel density analysis (Fig.2B) show an interestingly different story: with the commonest species growing increasingly common (at least in relative terms), and the rarest ones increasingly rare – in both terrestrial and freshwater plots. Thus while the number of species in the commonest category may be declining, the abundance of the remaining common species appears to be growing. This trend towards increasing community dominance by the few hyper-abundant species has been noted elsewhere (e.g. Biesmeijer et al 2006, *Nature* 313:351-353), but usually in analyses based on haphazardly collected biodiversity records – where such effects may be artefacts of shifting recorder behavior. Finding it here, in standardized sampling data, is a potentially important result. Figure 2B gets very little attention in the text (a single sentence, lines 132-133), but the apparent steepening of the SAD over time suggests that there is increasing inequality in abundances (rather than growing equality as suggested in the text). The fact that both the X and Y axes of Fig 2B are normalized makes the actual trends in abundance hard to assess. Moreover, as the abundance classes are normalized to the commonest species' abundance (if I understand line 413 and Extended data figure S1 correctly), categories themselves may shift as the commonest species grows increasingly dominant. If so, a common species whose abundance did not shift over time may nonetheless be demoted to a lower bin if the commonest species' abundance grows. It would help to give some thought to whether there's a way to meaningfully display species' abundance trends in a manner that is a bit more transparent. There is something interesting going on here: strongly significant shifts in the number of species in these different abundance bins (which the authors take to be indicative of growing evenness), and yet a steepening of the SAD suggesting increasingly dominance by those increasingly few common species, and increasingly subordinate rare

species(indicative of growing UNEvenness). The authors need to give more thought to interpreting these interesting shifts in the round.

(d) My conclusions in the previous section are constrained by my incomplete understanding of what the researchers actually did in this section. For example, it isn't clear (to me at least) what "in relation to the maximum value observed in the plot" (line 413) means. Is it the maximum value for each species (for sites sampled multiple times per year), or the most abundant species in each sample from the plot, or the most abundant species recorded in that plot in that year, or perhaps the most abundant species recorded from the plot over the course of the entire time-series? Conversely, the explanation of the Kernel density estimate (lines 420-428) is extremely detailed, but hard to understand.

(e) Concerning the data sets: the original van Klink paper (2020) did not include the 22 US LTER datasets presented by Crossley et al. (2020 NEE 4:1368-), and they don't appear to be listed among the new datasets added here. While inclusion of these time series would further strengthen the North American slant in the dataset, many of these appear to be relevant and of high quality. Is there a reason for this exclusion?

(f) Finally: a minor editorial point: surely the apostrophe in the paper's title is misplaced? It should be a plural possessive (assemblages'), as many different assemblages are involved. There are other indications of sloppy editing, e.g. presenting the first author's names differently in refs 13 and 18.

But I quibble. Overall, I was impressed with the ambition of this paper, but I'm not yet sure that they have made a water-tight case for some of their main findings. Until they can do so, I think it would be premature to consider publishing this work.

Author Rebuttals to Initial Comments:

Response to reviewers Van Klink et al MS 2022-06-08651

Reviewer #1

The manuscript presents a global synthesis of 171 long-term (>10yr) time series of insect abundance change in terrestrial and freshwater systems. The 'global insect decline' theme is extremely topical and important, and this manuscript represents an in-depth analysis of an exceptional dataset.

I must admit, though, that I really struggled to see the major new advance that the authors were trying to put forward.

Yes, the database has been expanded from the 148 studies in the authors' previous syntheses (Refs 13,18) to 171 studies here, through the use of additional search parameters (and publication languages). This is important, but not a major advance.

Yes, it is important that there were contrasting biodiversity trends identified between freshwater and terrestrial systems, but this was a key conclusion of the authors' earlier work (Ref 13).

Yes, it is important to try and break down these assemblage-level responses into contrasting responses across common vs rare species, and I like the general approach of looking at whether different biodiversity metrics (richness, evenness, dominance etc) are 'decoupled' due to idiosyncratic species responses in different systems. However, the concept that metrics can be 'decoupled' in the face of perturbation is not new (eg Ref 8), and some of the authors are involved with the more extensive study of this in the biorxiv pre-print by Blowes et al. (Ref 7).

Actually, I completely agree with the one of the Blowes et al conclusions that it is important to look beyond overall measures of 'abundance change' or 'richness change' because these can mask important (potentially contrasting) underlying processes. This is certainly what the current manuscript is trying to achieve, but the final outcome felt much more like an affirmation of concepts and trends that were already known, rather than anything that was really unexpected (to me).

The question is whether the headline result, of steeper trend slopes of decline in the number of species in the dominant abundance-class is sufficiently noteworthy. Refs 23 onwards give many specific examples of common/dominant species declining in abundance through time, but perhaps the generalization of this result for a compilation of 171 studies carries more weight? I am of two minds about this. It is a nice pattern (ie Fig 2, notably), but it is pretty much just presented as that; a nice pattern in the data, with no deeper insight about the processes or drivers. So, in a sense, the cited studies that give specific findings about decline/increase in individual study systems for individual species, are more informative in many ways. For a synthesis like this I would have expected some broader insight that could not be gained from the component studies, and I felt that was lacking here.

RESPONSE: We believe that the persistent declines of the formerly most abundant species at most sites (as now also shown by our additional analysis) is a new insight, and an important explanation for the insect decline phenomenon that has so far mostly been overlooked - . Loss of common species has been reported for other organism groups but never so comprehensively tested in insects. We believe this pattern is indicative of the scale of drivers of decline and the conservation action needed to reverse declines: a conservation focus on rare species is insufficient.

To better illustrate the implications of rare and common species loss, we have now developed three hypothetical scenarios, which we use to unite the trends in different biodiversity metrics. Our results disentangle some of the seemingly contrasting findings in the literature (e.g. abundance declines, and much smaller (if any) richness declines, and variable (centering on zero) population trends for insects (Pilotto et al 2020, Crossley et al 2020) and other organisms (e.g. Dornelas et al 2019)).

There are a few reported examples of declines of specific formerly abundant and relatively charismatic species like Monarchs but these cannot be taken as evidence that this is a general phenomenon also among less charismatic insects. Our synthesis is highly indicative that this is a large-scale problem. Moreover, by analyzing assemblage datasets, we can show that the declines of formerly common species is not compensated by increases of formerly more rare species.

Our results show that declines happen for a wide variety of species, and we believe it's unlikely that there is one single driver or life history trait responsible for this. The wide variety of data in our study do not allow for an analysis of drivers, as drivers probably differ vastly across sites, regions and taxa,, and relevant data (e.g., on land-management practices, including pesticides/pollution) are often not available at the global scale. There are ongoing projects that will be able to address this in the future, but at present, the data are just not sufficient.

In addition, I had quite a number of problems with the ways in which the analyses were conducted, and as a consequence I did not find the general conclusions overly compelling, or robustly proven.

Stepping through the analytical issues, using the Figures as focal points:

Fig 1 looks lovely at face value, but is disappointingly uninformative on deeper inspection. As the authors note themselves Fig 1b just reflects a trivial sampling process in which richness increases with sample abundance. This phenomenon is so well known and has little bearing on biological interpretations of the data, so I can't see any justification why Fig 1b should be in the main text, rather than in the extended data figures or supplement.

RESPONSE: Figure 1b has now been removed from the manuscript. The reviewer is right that, especially for the terrestrial data, it reflects (on average) a sampling process. This was not the case for the freshwater data, but these have been removed now.

The main panel to Fig 1a presents abundance and richness trends for freshwater and terrestrial studies in the upper two frequency distributions, which is fine. The numbers to the right of the plots seem to represent number of studies/sites for each metric, and that raised an immediate red flag about the comparison of metrics based on differing numbers/identities of studies compiled. Across metrics, Ext Dat Table S1 and Fig 1a seem to suggest that there

might only be 54 terrestrial studies and 20 freshwater studies (or fewer) out of 171 that would have all four comparative metrics of abundance, richness, rarefied richness and evenness for the same suite of studies (?). Given the earlier preamble in the Introduction about 'decoupling' of metrics, it would seem important to identify the subset of studies in which it is actually possible to tease apart *all* the metrics, and perhaps present that as a sensitivity test in the supplement.

RESPONSE: The reviewer is right that there is a difference in the number of datasets and sites available for some of the metrics (with much more data available for richness and total abundance than for the metrics that require full species-level data), and we apologize for not being clearer with this. We have now added the number of available sites with species level data to the abstract.

In this new version, we now also show the estimates for richness and abundance based on only the datasets that have full community composition (dashed lines). This shows that for the terrestrial data, the estimates for the full dataset and the reduced dataset are very similar.

For the remainder of Fig 1a (the rarefied richness and evenness frequency distributions), I don't think the metrics used are really fit for purpose.

In the Methods section ('Univariate metrics') the authors indicate that rarefied richness was interpolated at the lowest total abundance observed in any year in the timeseries (except when lower than 10 individuals, in which case rarefied richness was extrapolated to a 10 individual sample). Rarefying in this way throws away most of the information in the data, and forces a comparison among studies at potentially very low sample abundances, right near the steeply-ascending basal portion of the species accumulation curve. We don't know how serious a problem this is, because we are not given the information about the abundance level at which the rarefaction values were calculated for each study. In most cases, though, there will be almost no power to discriminate richness trends using this rarefaction approach (hence, no surprise that there appears to be little resolution in rarefied richness trends).

RESPONSE: We have now additionally calculated 'coverage-based richness', as suggested by this reviewer. This showed a very similar trend as rarefied richness does. For transparency, we have retained both metrics in figure 1.

Later in the methods (under 'Preparation of raw data'), the authors use a different approach to standardisation of sampling across years, which is to subsample in order to equalise sampling effort (eg number of trap samples), rather than equalising sample abundance (as in the case of rarefaction above). They go on to describe some examples of situations in which the data were wrangled into shape based on standardising sample effort, but this very much sounds like a subjective set of approaches that will be inconsistent across studies and analyses.

I would suggest changing to a more objective approach for all these types of standardisation (rather than by sample abundance or by sampling effort), and instead standardise objectively by sample coverage (eg see papers by Anne Chao, and the recent Roswell et al 2021 paper in *Oikos*). Sample coverage intrinsically recognises that small sample abundances and/or low sample sizes could actually accumulate a better 'coverage' of a small total fauna than large samples taken from a much larger total fauna (so, it's not only about the size of the sample, but about the degree to which it captures the species pool in a representative manner). There are R packages like 'iNEXT' or 'meanrarity' that could interpolate/extrapolate

to a common sample coverage value (like 0.8 or whatever), which would be more objective and use more of the available data (in a more consistent manner).

RESPONSE: As the reviewer states, changes in biodiversity can only be assessed if sampling is comparable. The reviewer suggests that this can be achieved using coverage-based biodiversity metrics.

Unfortunately, we do not agree that this is possible given these approaches and that coverage-based analyses, while often quite useful, do not provide the complete control over sampling effect as is often implied. Specifically, this ignores the complex sampling histories of most of the datasets used here, where sampling effort was not equal across years, and more importantly, not across seasons. As an example: if in a year three samples were taken, 2 in spring and 1 in late summer, this could give a very different coverage based richness than when 1 sample was taken in each season in another year, as the spring species are overrepresented.

In other words, our approach for equalizing sampling effort across years by accounting for seasonal variation in sampling effort is still needed even when coverage based diversity metrics will be calculated. We understand that our approach may seem subjective, but this was needed to maximize use of the available data, rather than discarding data that was not precisely matched seasonally.

Nevertheless, we have now added coverage based richness (0.8 coverage) using our standardized data in order to develop a more complete analysis of metrics.

Next, for evenness in Fig 1a, I have even bigger concerns. The authors describe their evenness measure as ENS-PIE “Hurlbert’s probability of interspecific encounter (= inverted Simpson index), converted to the effective number of species”. The authors also state in the methods “We calculated a number of biodiversity change metrics based on the framework of Hill-numbers (Ref 21). This framework unifies different metrics of biodiversity, evenness and compositional change”.

This set of statements does not make sense to me, as written.

First, Hurlbert’s PIE is related to the *complement* of Simpson’s index (ie $1 - \sum(p^2)$), not to inverse Simpson ($1/\sum(p^2)$). Moreover, the complement of Simpson’s index is otherwise known as the Gini-Simpson diversity index, and is not (in its own right) a measure of evenness. It is a measure of diversity. There is also an important distinction that Hurlbert’s PIE is calculated by sampling *without replacement* whereas the complement of Simpson’s is calculated by sampling *with replacement* (and this makes a big difference when the sampling universe is finite, such as in many of the small-sample cases that the authors describe in their data). Then, the conversion to ‘effective number of species’ (ENS) is presumably what the authors imply by their use of the Hill series framework [ie inverse Simpson, $1/\sum(p^2)$, is indeed the effective number of species for the Simpson measure]. Of course, this ENS measure is also based on sampling with replacement, and crucially is a diversity measure too, not an evenness measure. Of all the Hill series ‘numbers equivalents’ of diversity measures that the authors could have chosen, it is also the one that is most highly influenced by the dominant species in the assemblage.

RESPONSE: We thank the reviewer for these comments and corrections. To avoid confusion, we have now removed the statement that we use the Hill-number framework, even though the majority of the metrics we use fall in this framework. We have added the Shannon index (ENS) and coverage based richness, have renamed ENS PIE to the inverse Simpson index (which is the way we actually calculated it), and have added a ‘real’ evenness metric (see below).

I think this is a major problem. If the authors claim to be using the Hill series framework, then a quick read of Hill 1973 would show that Hill proposed evenness measures that are a ratio of any two ENS ‘numbers equivalents’, such as $E_{1,0}$, or $E_{2,1}$ (the latter for example being the ratio of inverse Simpson divided by exponential Shannon). Classics like Smith & Wilson 1996 *A Consumer's Guide to Evenness Indices*, *Oikos*, provide useful information on this, and alternative evenness measures (they advocate Evar, for instance). There are many different options for a genuine evenness measure that could be adopted, rather than the ‘diversity’ measure that the authors use here, masquerading as evenness.

Of course, the Hill series ‘numbers equivalents’ of diversity are useful to use in their own right (as diversity indices), but I would suggest not just using inverse Simpson (the most affected by dominant species), but rather the range of metrics right across the Hill series that provide differential weight to rare vs common species. Read the Roswell paper for additional insight on this.

RESPONSE: We have now renamed what we previously called ‘evenness’ back to the Simpson index, and have calculated a ‘true’ evenness metric, as the reviewer suggested: we now present evenness as the ratio between the Simpson index ($q=2$) and species richness ($q=0$). This index is among those recommended by Smith and Wilson (1996, called $E_{1/D}$ in that paper). We have also experimented with Evar and Pielou’s index: In our simulations, these indices showed the same qualitative patterns as q_2/q_0 (i.e. consistently increasing, stable or declining in the three scenarios, see below), and hence are largely interchangeable.

Due to convergence problems in our statistical models with Evar and Pielou, we retained q_2/q_0 (last panel in the graph) as our evenness metric.

Figure 2 – trend slopes by abundance percentile class in the species-abundance distribution: This is the only really important figure in the manuscript, and the one that seems to have the fewest analytical problems associated with it.

The authors have done their due diligence in appropriate sensitivity tests of different ways of slicing up the SAD distribution (either 20% abundance intervals, as shown in Fig 2, or equal numbers of species per quartile in Fig. S6), and the kernel density estimate to show changes in the shape of the SAD in Fig 2b is effective.

I do wonder about a couple of things stemming from ‘relativising’ commonness vs rarity at the plot level. One is that the same study could have multiple plot/site locations, and relativising a species as ‘common’ (ie 80-100% abundance interval) in plot 1, where all species have low abundances because the site is disturbed, seems quite different than the same species being designated as ‘common’ in an adjacent natural site Plot 2 where total assemblage size is much larger. Absolute abundances would be very different between

Plots 1 and 2 for this species, but in both cases the SAD relativisation would potentially deem them equally common.

RESPONSE: This is correct, the species would be assigned 'most abundant' in both plots.

The authors will no doubt argue that this doesn't matter because it is about the relative trend through time at each plot (separately). However, it does matter because the authors key conclusion is that the *number of species* in the 'common' 80-100% SAD interval is declining through time, but that is definitely not the same thing as designating an overall classification of 'common' vs 'rare' and then investigating the *abundance* trends of the common species to show that their rate of decline in abundance is steeper than for species in rarer abundance classes. In the original 171 studies, authors would have used the absolute abundance data from multiple plots (eg disturbed vs undisturbed sites etc) to infer abundance trends through time for each species, and my gut feeling is that the authors approach in this manuscript showing richness changes per SAD abundance class would not mirror abundance trends through time estimated at the species level.

RESPONSE: We thank the reviewer for the support of our SAD analysis. Indeed, we were only interested in local scale changes. For our questions regarding insect biodiversity change at local scales, the local SAD was most appropriate, and it should not be confused with analyses of species trends in relation to their range or occupancy at larger geographic scales (in other words, we ignore the geographic range axis of the Rabinowitz classification). Please also note that we have avoided the word 'common' throughout the manuscript, to avoid confusion. We now refer only to locally 'abundant' and rare species (having also removed the word 'dominant' from the manuscript, as this can also lead to confusion (often interpreted as 'regionally widespread and abundant' or having a strong impact on other species (see Avolio et al 2019 New Phytologist)).

We agree that studying the trends of species in relation to other definitions of commonness is also interesting, for example occupancy in a meta-community or by global distribution metrics. Unfortunately, this was not possible for many of our datasets, as many have only one site, whereas in other datasets the environmental heterogeneity among plots is so that some species have zero abundance in some plots simply because the habitat is unsuitable. This would be an inappropriate way of assigning overall commonness and rarity, because of unclear representation of habitats..

For an analysis of metacommunity occupancy trends in relation to species' ranges, we refer to Xu et al 2023 Nature Communications: <https://www.nature.com/articles/s41467-023-37127-2>

Lastly on this point, I would also say that most readers would be misled by the title of the manuscript, "Widespread declines of dominant insect species", which implies that the authors are reporting abundance declines of common species, which is a different thing than what they show in Fig 2 (declining richness of common species at a site).

RESPONSE: We have now included an additional analysis at the population level, to test whether or not, on average, the most abundant species in each plot decline more or less than the rarer species.

Our analysis shows that indeed the locally most dominant species are declining more than the rare ones, and this is largely independent of the reference period for classifying dominance. We think that, given the new analyses, the title, with slight adjustments, is thus justified.

Figure 3 – species turnover

I was quite surprised that such a crude approach was used for species turnover through time. At one level it is good to contrast the qualitative presence-absence metric with a quantitative abundance metric, but the Morisita-Horn metric is notable for being strongly influenced by the most abundant species in the assemblage. In this sense, yes it is a match to the inverse-Simpson, but that is not a good thing in terms of a conservative approach to assessing whether changes in dominant species are driving assemblage-wide restructuring. The use of Morisita-Horn represents an undue bias toward the intended outcome. Morisita-Horn also does not handle joint absence similarities as well as some other dissimilarity metrics either.

As a second major problem, the turnover values are calculated as repeated pairwise dissimilarities to the single year at the start of each time series. It should be self-evident from the recent insect decline literature that this will create a potential bias akin to a 'baseline effect'. Consider two examples: first, if the baseline year had an aberrantly dissimilar assemblage (visualised in an ordination plot this would be a point very far from the overall plot centroid) then all subsequent pairwise dissimilarities would be high, making it impossible to discern any trend in assemblage composition across the remaining years (whether one existed or not); second, imagine baseline assemblage composition close to the plot centroid value in an ordination, there could easily be a false appearance of gradually increasing pairwise dissimilarities through time, even if these were distributed in random vector directions each year (eg alternately increasing/decreasing along axes 1 and 2 of an ordination in different years, but not tending directionally along one ordination vector).

There is no way of telling what the magnitude of this problem is, but I would have no confidence whatsoever in the trends represented in Figure 3. There are many ordination methods for testing predictor effects on compositional trends in multivariate space. Exploring one or more of these approaches, along with a judicious left-censoring of time series to explore potential baseline effects would be well advised.

RESPONSE: The reviewer is right that an analysis of beta-diversity requires more thought than we put in originally, and we have removed this analysis in favor of the population level analysis. Our approach to beta diversity was rather crude, but in our opinion there are no other methods available that can be applied to all the data we have, as many of these datasets are sparsely populated. Although we have not plotted and inspected turnover results at every single site, the sites that we did look at consistently showed a gradual increase in beta diversity compared to the baseline. Such a turnover is exactly what is to be expected, even in the absence of environmental changes (see Dornelas et al 2014 or Hubbell 2001). The scenarios of aberrant baseline years sketched by the reviewer are both possible, but probably rare in our data.

Referee #2 (Remarks to the Author):

The past decades, the biodiversity crisis has been frequently alerted to. Importantly, large scale data analyses have provided deeper insights into the rate and processes of biodiversity changes on our planet. The sometimes dramatic declines of insect abundance and its consequences for ecosystems functioning have been strongly quantified in more recent years. Metrics that have typically been presented to quantify biodiversity loss of insects have been total biomass, abundance, taxonomic richness as well as species turnover. Several studies have provided insights into how these metrics differ for realms, geographic areas or specific species groups. In their current study, van Klink and co-authors

explore whether aquatic and terrestrial insect communities differ in biodiversity changes and how the rarity of insect species in communities is characterising the trends of biodiversity changes in the two realms differently. The study reveals that declines in dominant species in both realms occur, but that in aquatic systems (and not terrestrial) these are compensated by a rise in numbers of rarer species. These results are obtained by building an extensive database of reported studies that present longer term changes in biodiversity of their study sites. Overall the study presents a rigid data analysis, although some methodology could be presented in more detail (see specific comments below). The graphical presentation of the data transfers a clear message and is easily interpretable. I appreciate the use of alternative bins and approaches to show that the results are not sensitive to more arbitrary decisions in data analysis.

My overall assessment of the study is that the study connects to perhaps the most important crisis we are currently facing, that of biodiversity decline and its consequences to ecosystems functioning, and the cascade in numerous societal consequences. It presents clear conclusions with rigid data analysis. At the same time, I have reservations in my enthusiasm for this study. These include, i) novelty, in terms of extending our insights compared to published work, ii) overreaching of conclusions by the terminology/wording used, iii) unclarity of some sections in the main text and methods that make the manuscript less accessible to a broader readership (at least including myself).

RESPONSE: Thanks for your positive feedback

i) The authors heavily built on their wonderful recent study in *Science* (2020; 368, 417-420), where they analyse a similar dataset for biodiversity metrics. Their previous work revealed that patterns in insect decline in terms of abundance differ for aquatic and terrestrial ecosystems. The same message is prominently part of the current manuscript, but deepened with analysis on metrics such as richness, evenness and species turnover. The responses of these metrics for aquatic and terrestrial realms show a pattern that is similar to work by Blowes et al (2019; *Science* 366, 339-345). Species turnover was found to be a dominant factor in biodiversity trends and that differences in turnover contributed to pattern differences for aquatic and terrestrial systems. Pilotto et al. (*Nature Communications*; 2020, 11:3486) similarly identified species richness trends that differ for terrestrial and aquatic systems and that taxonomic groups differ in trends of biodiversity metrics. Also this study shows the difference in responses of terrestrial and aquatic insect/invertebrate communities. In a more theoretical paper by Blowes et al. (*Ecology*, 2022, in press) the interactions between metrics are evaluated for their characterisation of biodiversity trends. The authors do acknowledge most of these and other studies in their manuscript. Although this new manuscript presents new analysis of common versus rare species, it also tells much of what other studies have shown. I do certainly appreciate the overall work, but also had expected a more significant novel message when placing this in context of *Nature*.

RESPONSE: We agree that some of the analyses presented here are similar to those in other studies listed here. The major advance in understanding is the different trends by rare and locally abundant species, which is also reflected by our deepened analysis of the Shannon and Simpson indices, as well as evenness. It should also be noted that we find a slightly declining richness trend, countering Pilotto et al and Crossley et al. We think that our findings indicating that insect declines are overall driven by declines of the most common species has wide repercussions for the current debate about what is needed to tackle insect loss and therefore of sufficiently broad appeal to scientists, policy-makers and conservation organizations for a journal such as *Nature*.

ii) Some of the statements, conclusion and title oversell the findings of the manuscript. By line number suggestions, I highlight some of these sentences below. Some of the strong statements are due to the choice of words. The term community structure is used by many

ecologists to particularly describe species interactions, food web relationships and how assemblies form in specific areas, on shared resources etc. What the data is presenting are metrics of species observations, not about structure in terms of species relationships/interactions. When reading the title, I was expecting to see data on how loss of dominant species alters the interaction structure of an assembly. In the abstract the statement “ abundance changes would lead to predictable changes in species numbers and community structure” , speaks differently to people working on keystone species, density mediated processes, priority/legacy effects in communities, as well as the role of biodiversity to stabilise communities (through structure). In these research fields it is a given that a change in abundance of one species may have very unpredictable consequences for community structure. I suggest to use the term biodiversity metric throughout the manuscript.

We have extensively revised the text and changed the title to further prevent such confusion. The term ‘community structure’ no longer occurs in the manuscript.

iii) In the introduction of methods in the main text as well as the methods section, some statements are technical and difficult to judge what it means for the analyses. For example, a) what does it mean for the reliability of your analyses when there is strong evidence with a 95% confidential interval, but weak evidence with 90 or 80%. Here you could help the reader by explaining what part of the story changes when using a different confidence interval.

RESPONSE: We apologize for the lack of clarity. These different credible intervals do not say anything about the reliability of our analysis, but are an indication of how certain we are of a given mean estimate. The intervals mean that we are 80 or 95% certain that the mean estimates of our data lie between these borders. Hence, we can infer the strength of evidence, as is done in IPCC and IPBES reports: strong evidence means that we are over 95% certain that a trend is larger or smaller than zero. ‘Weak’ evidence means that we’re only 80% certain, so that there is a somewhat larger chance (20%) that there is actually no trend in our data

b) it is unclear to me how you can characterise abundance trends over time when you use a variable starting point in time, based on the highest abundance in the full time series. The explanation in the main text (Page 4, final paragraph) is too technical for me to understand how species are treated that have initial decline and recovery in later time points.

Because of the variability in the data, we can only use the simplest models, namely (log)linear (a certain % change per year) models. We know and acknowledge that the trends of all these metrics are in reality often not linear (i.e. may go up and down over time), but with the available datasets, many of which have only 2 or 3 time points, fitting non-linear models is not possible. Therefore we report only the yearly change estimate (see below for more explanation). At least part of the non linearity will be accounted for by the auto-regression term, but we focus our analyses only on the mean long-term trend i.e., average change from one year to the next across all years with data. In these terms, our analysis is similar to other recent synthesis of biodiversity change (e.g., Dornelas et al. 2014).

c) in the methods I could not find how the authors calculated the rate by year, which is the central value used in all metrics. In the methods section, it would have helped me to make the rate by year very visible (perhaps even in a header).

The yearly change value is the estimate of the ‘year’ term from the linear models. Now explained in L892-895 of the method section. This is the same as what we did in our previous work (Van Klink et al. 2020).

'By focusing only on the temporal slopes, we could account for the different scales of measurement (from 1 to thousands of individuals, and up to hundreds of species per year). The coefficient of the 'Year' variable (the temporal slope) can be back-transformed to the percentage change per year, and we report both in all graphs.'

By line number

Line 24: This statement is too strong. How abundance trends reflect changes in other biodiversity aspects is presented in many other papers that you cite, and thus not unknown.

RESPONSE: rephrased to 'less clear-cut' to indicate that there is disagreement even among some of the larger studies.

Line 25: Here "community structure" is not the best term to use as at least part of the readers of your paper would disagree with the statement.

RESPONSE: we have removed the phrase 'community structure' from the paper

Line 38: there are also many ecological studies that show that dominance of a species is not a good predictor of how it affects community structure / ecosystem function.

RESPONSE: We agree that for rare species it is often unknown and sometimes surprising how much of an effect they can have on the ecosystem, but for abundant species it is to be expected that they are more important as food for higher trophic levels and will consume more resources, and thus have an impact on the system. This is excluding any potential role as ecosystem engineers. We now added some references to support our statement as well as the nuances.

Line 55: here the term community structure should be avoided

RESPONSE: Done

Line 75-77: Here I suggest to make the more technical statement into a statement of what it means for the confidence of your ecological statements.

RESPONSE: we have reworded this as: 'Our models present the yearly change estimates and the 80%, 90% and 95% credible intervals (CI) of these estimates. Following^{13,19}, we interpreted any 95% CI that did not overlap zero as strong evidence for a directional trend (i.e we are over 97.5% certain that there is a directional trend in our data), while a CI overlapping zero between the 90 and 95% CI's was interpreted as moderate evidence, and an overlap between the 90% and 80% CI's was interpreted as weak evidence (i.e. a 10% chance that the actual mean of the data is zero).'

Line 82: This sentence seems incomplete

'These results were robust to the exclusion of very long and very short time series (Extended Data Fig. S3), but restricting the maximum number of sites per study (to 50, 20 or 10) led to progressively more positive terrestrial richness trends in both realm, indicating that larger studies tended to have lower mean estimates (Extended Data Fig. S4).'

RESPONSE: reworded as: ' Furthermore, these trends were robust to the exclusion of very long and very short time series (Extended Data Fig. S2). However, studies with a large number of sites tended to have stronger abundance and richness declines, as restricting the number of sites per study (to 50, 20 and 10) led to more positive trends in all metrics (Extended Data Fig. 3).'

Line 83: The second part of the same sentence is not clear on what is meant with " lower mean estimates" of which parameter? Larger studies could be defined for number of sites or time series.

REPONSE: We have changed this sentence: 'but restricting the maximum number of sites per study (to 50, 20 or 10) led to progressively more positive richness trends, indicating that studies with more sites tended to have stronger species richness declines (Extended Data Fig. S4).'

Line 131: here you could use rare / abundant species also in statements such as " strong evidence for declines of the upper 10%"

RESPONSE: Sentence has been removed.

Line 138: Here the richness trend of terrestrial species is considered constant. This however depends on the confidence interval used. An overall low yearly rate may still be a significant decline over a long time period.

REPONSE: The reviewer is right that even a small rate of change can be quite large over multiple decades. However, the confidence that this change is real will not change over time. Regardless, the sentence has been removed during revision.

Line 142-151: This section is written very technical and it is not easy for the reader to identify how your method provides an accurate view on the yearly rate of change in a community metric.

RESPONSE: We have rewritten this section for clarity on the problem of RtM effects. For the technical details we refer to the Methods section, as there is no space for this in the main text.

'A common pitfall when relating population trends to their initial values is detecting false trends due to a 'regression to the mean' effect². That is, if an extreme value is found at the start of a time series due to stochasticity in population dynamics or sampling variation, the values for the following years are likely to be lower, which would lead to falsely detecting a declining slope over time. Hence, the species with highest starting values are likely to show the strongest declines and species with the lowest starting values the strongest increases, simply by returning to their long-term averages. To avoid this, we corrected the detected trends for each abundance interval by adding an expected regression-to-the-mean estimate calculated from the most robust time series (Extended Data Fig 6).'

Line 170: remove first full stop before reference to extended data

RESPONSE: Sentence has been removed

Line 187: structure should be replaced by biodiversity metrics

REPOSNE: we have removed all instances of 'community structure'

Line 187-205: The discussion here connects well to literature and also hints that other studies have shown that especially the abundant species are in decline. This makes me conclude more strongly that the insights presented are less novel. This certainly does not make your study less valuable, but could indicate that the finding is not of the level of novelty searched by the journal.

REPOSNE: It is true that there are case studies that have shown that some formerly abundant species have declined and a number opinion pieces on the topic. However, most of the case studies mentioned are no comparisons among species, but simply observed declines of previously abundant species. Also there has been no synthesis of this, and it has not been connected to insect declines. We now better emphasize our novelty on lines 254-259.

Referee #3 (Remarks to the Author):

This paper is a successor to a highly influential recent paper in Science (2020 368:417-420), which presented a meta-analysis of trends in insect abundance, biomass and species richness in terrestrial and freshwater communities. This new manuscript extends the dataset (adding 23 new studies), revises one of the earlier findings (with declines in terrestrial species richness no longer being well supported),

REPOSNE: With all due respect, this is a misunderstanding. In our 2020 paper, we did not test changes in species richness, and the richness component is one of the new aspects in this paper.

and explores additional aspects of biodiversity: evenness, abundance structure and temporal turnover. The manuscript is reasonably well written (especially the introductory section, which lays out the issues skillfully), and it reports some striking new results: an increased evenness in species' abundances over time in both terrestrial and freshwater arthropod communities, which the authors link to a decline in the number of very abundant species, while the number of rare species holds steady or increases. If those findings prove to be robust, the paper could make a substantial impact, contributing to the ongoing policy and public debate on this issue.

However, in its currently form I would have some qualms about recommending this manuscript for publication (in Nature or elsewhere).

(a) To begin with a relatively minor matter: the order of the reporting of main results (lines 78-11) seems odd. The natural order is to go from (i) summed abundance to (ii) species richness to (iii) rarified richness, to (iv) evenness – and indeed that is the order used in the Figures (Figs 1, S3,S4, S5). The manuscript moves the species richness result first, and (understandably) gives it disproportionate emphasis, presumably because it differs from the result published earlier (with terrestrial richness declines no longer deemed significant). However, doing so disrupts the flow of the argument in my opinion.

REPOSNE: We have changed the order of the results discussed to follow the order in Fig 1 (which we have also adjusted)

(b) More fundamentally, the index used for measuring evenness (Hurlbert's PIE index, which is equivalent to the finite sample form of Simpson's 1-D) is sensitive to both richness and evenness. It's a fine index, but it is not properly speaking an evenness index (although others have sometimes called it one, e.g. Wall et al. 2018, *Biodiv & Conserv* 27 395-415), but rather a measure intermediate between a pure richness index and a pure evenness measure. As species richness is already incorporated into the analysis (both in raw and rarified forms), it would be preferable to use an evenness index that was independent of richness. The authors express a desire to use an index from the Hill family of indices, and it's true that a variant of PIE (Simpson's 1/D, which I believe is equivalent to the "expected number of species" variant used here), is a member of that family of indices with an exponent of 2 (and thus more evenness-weighted than exp(Shannon) with exponent 1),

RESPONSE: The reviewer is right, and this has also been pointed out by Reviewer #1. We have now reworded several sections, and added some more metrics (the Shannon index (exponential), coverage based richness, and have added a 'true' evenness index (see below, and our response to Reviewer #1).

but there is a proper evenness index in the Hill family: the Berger-Parker index, with exponent of infinity. There are far better evenness indices than Berger-Parker (see e.g. Smith & Wilson 1996, *Oikos* 76:70-82; Beisel et al. 2003, *Internat. Rev. Hydrobiol.* 88: 3-15; Tuomisto 2012, *Oikos* 121: 1203-1218; and many others). There are even evenness numbers derived from Simpson's index: Hill himself published one (Hill 1997, *Oikos* 79: 413-416), and a more widely used variant was proposed by Krebs (1989: *Ecological Methodology*): $(1/D)/S$. PIE might be included as a general-purpose diversity index (and one that is relatively insensitive to sample size), but if the goal is to reflect evenness per se, the index is a poor choice.

REPOSE: As a more appropriate evenness index, we have now included $(1/D)/S$, which is one of the indices recommended by Smith & Wilson (1996). For more details we refer to our response to Reviewer #1

(c) The section on abundance structure is also in need of refinement, and of better explanation. The authors devote most of their attention (lines 119-131) to the number of species in equal-sized abundance categories, and find a decline in the number of the commonest species, and an increase in the number of rare ones. But they present this in misleading terms: phrases like "Widespread declines of dominant species" (title), "declines of (formerly) dominant species" (abstract), "the most abundant species disproportionately declining" and the "rare species disproportionately increasing" (lines 119-120) sounds like they indicate shifts in the abundance of these species, rather than in the numbers of species in each category. These categories are themselves not fixed, but rather scaled relative to the commonest species in each dataset. Meanwhile, the Species-Abundance distributions derived from the kernel density analysis (Fig.2B) show an interestingly different story: with the commonest species growing increasingly common (at least in relative terms), and the rarest ones increasingly rare – in both terrestrial and freshwater plots. Thus while the number of species in the commonest category may be declining, the abundance of the remaining common species appears to be growing.

RESPONSE: This is an important observation. We have investigated this further, and must conclude that, unfortunately, the kernel density approach as applied here is uninformative regarding changes in community composition (for more explanation on this see the end of this document).

We have now removed the KDE analysis, as well as the analysis of beta-diversity, and instead analyse population trends at the species-level. This is a more direct test of our

previously asserted pattern of stronger declines for formerly abundant species. This assertion was confirmed, hence we have kept the title.

This trend towards increasing community dominance by the few hyper-abundant species has been noted elsewhere (e.g. Biesmeijer et al 2006, Nature 313:351-353), but usually in analyses based on haphazardly collected biodiversity records – where such effects may be artefacts of shifting recorder behavior. Finding it here, in standardized sampling data, is a potentially important result.

RESPONSE: We agree. We also now more formally test this by calculating population trends for different abundance classes. This was insightful, as it showed that the locally most abundant species indeed decline strongest.

It should be noted that this analysis differs from Biesmeijer et al's, as our test only refers to local communities, and assesses commonness on a local basis. As stated in our response to reviewer #1, we were interested in local changes to explain local insect declines, and not in overall qualifications of species' range size or occupancy patterns. We think this is also an interesting question but has already been asked e.g., Xu et al. 2023.

Our results seem to contrast Biesmeijer's, as the formerly locally most abundant species have declined most, and the rarer species on average less (or were replaced). Nevertheless, formerly abundant species might still be the most widespread and/or locally abundant, but we find no evidence for more skewed local communities (i.e. very dominated by few species), but rather for more equal communities.

Figure 2B gets very little attention in the text (a single sentence, lines 132-133), but the apparent steepening of the SAD over time suggests that there is increasing inequality in abundances (rather than growing equality as suggested in the text). The fact that both the X and Y axes of Fig 2B are normalized makes the actual trends in abundance hard to assess. Moreover, as the abundance classes in are normalized to the commonest species' abundance (if I understand line 413 and Extended data figure S1 correctly), categories themselves may shift as the commonest species grows increasingly dominant. If so, a common species whose abundance did not shift over time may nonetheless be demoted to a lower bin if the commonest species' abundance grows.

REPNSE: This is a misunderstanding, and likely a result of inadequate explanation on our part. A species that does not change in abundance stays in the same bin at all times, because the bins are fixed relative to the highest observed value in the time series. Because this analysis is very hard to explain and not very diagnostic of changes to the community, we have removed it.

It would help to give some thought to whether there's a way to meaningfully display species' abundance trends in a manner that is a bit more transparent. There is something interesting going on here: strongly significant shifts in the number of species in these different abundance bins (which the authors take to be indicative of growing evenness), and yet a steepening of the SAD suggesting increasingly dominance by those increasingly few common species, and increasingly subordinate rare species(indicative of growing UNevenness). The authors need to give more thought to interpreting these interesting shifts in the round.

RESPONSE: This is now more formally tested in the models of population abundances, which provides more direct evidence that formerly common species have declined. Our new Box 1 shows how this connects to the SAD and other biodiversity metrics.

(d) My conclusions in the previous section are constrained by my incomplete understanding of what the researchers actually did in this section. For example, it isn't clear (to me at least)

what “in relation to the maximum value observed in the plot” (line 413) means. Is it the maximum value for each species (for sites sampled multiple times per year), or the most abundant species in each sample from the plot, or the most abundant species recorded in that plot in that year, or perhaps the most abundant species recorded from the plot over the course of the entire time-series? Conversely, the explanation of the Kernel density estimate (lines 420-428) is extremely detailed, but hard to understand.

RESPONSE: We have attempted to explain these methods better. The ‘maximum value observed in the plot’ refers to ‘most abundant species recorded from the plot over the course of the entire time-series’. We have added more explanation (L 727 ff.), and added more explanation in Extended Data Fig 4:

‘First, we created five equally sized intervals of the \log_{10} abundances (Extended Data Fig. 4b), between zero and the maximum abundance observed at that site over the course of the time series. In other words, the baseline for ‘locally abundant’ at a site is the highest abundance of any species recorded from the site over the course of the entire time-series.’

(e) Concerning the data sets: the original van Klink paper (2020) did not include the 22 US LTER datasets presented by Crossley et al. (2020 NEE 4:1368-), and they don’t appear to be listed among the new datasets added here. While inclusion of these time series would further strengthen the North American slant in the dataset, many of these appear to be relevant and of high quality. Is there a reason for this exclusion?

RESPONSE: This is not the case. Most of the datasets used by Crossley et al that fit our criteria were included in our analysis. Below, we list the 22 studies used by Crossley and indicate whether they were excluded from our analysis and for which reason. The only study used by Crossley that fit our criteria, but was not included is the Midwest Suction Network data. We were unaware that all data from this study were available in the supplement, and since it fits our criteria, we now include it in our analysis

Arctic	Arctic tundra Stream insects	included only for abundance
Baltimore	Urban Mosquitoes	Too short
Bonanza creek	Taiga Bark beetles	Too few species
	Aspen leaf miner	Too few species
Cedar Creek	Savannah/tallgrass prairie Arthropods	experimental
	Grasshoppers	included
Phoenix	Urban Ground arthropods	included
	Shrub arthropods	poor quality, no standardized sampling
Coweeta	Temperate deciduous forest Aquatic invertebrates	included only for abundance
Georgia Coastal Ecosystems	Salt marsh/estuary Crabs	non insect
	Grasshoppers	included
	Planthoppers	Too few species

Harvard Forest	Temperate deciduous forest Ants	included
	Ticks	Too few species
Hubbard Brook	Temperate deciduous forest Lepidoptera larvae	included only for abundance
Midwest farmland	Row crop agriculture Aphids	Now included
Konza Prairie	Tallgrass prairie Gall insects	too short
	Grasshoppers	included
North Temperate Lakes	Temperate lake Pelagic/benthic macroinvertebrates	included
	Crayfish	non insect
Sevilleta	Desert/grassland Grasshoppers	included
	Ground arthropods	poor quality. Sampling was unreliable.

(f) Finally: a minor editorial point: surely the apostrophe in the paper's title is misplaced? It should be a plural possessive (assemblages'), as many different assemblages are involved. There are other indications of sloppy editing, e.g. presenting the first author's names differently in refs 13 and 18.

RESPONSE: This has now been fixed.

But I quibble. Overall, I was impressed with the ambition of this paper, but I'm not yet sure that they have made a water-tight case for some of their main findings. Until they can do so, I think it would be premature to consider publishing this work.

More on the Kernel Density Estimations:

We calculated the KDE of the SAD for each of our three scenarios to see if it is diagnostic for the changes in the community:

As is clear from this graph, the three very contrasting scenarios give qualitatively fairly similar patterns. We believe that this may be because 0's are retained for the KD estimation over time, and because the KDEs always have the same area under the curve, and hence, when the total number of individuals declines (as in all our scenarios), the actual pattern of change is not straightforward to interpret. In addition, the adding of the area <0 to the positive part of the curve may distort the pattern as well, especially when there are 0's in the data, though we still believe that some correction is necessary, as the positive area of the curve will be diminished, as more of the curve moves below 0.

There may yet be a use for KDE for studying SADs, but it will require more thought regarding how to best do the analysis and how to interpret the results.

Reviewer Reports on the First Revision:

Referees' comments:

Referee #1 (Remarks to the Author):

Review of MS 2022-06-08651A Nature
Widespread declines of formerly abundant species drive insect loss
van Klink et al.
20th June 2023

I commented on an earlier version of this manuscript (as reviewer #1) and had quite a number of reservations about the work. The resubmitted version is an entirely different beast. All credit (<audible applause>) to the authors for taking the time to think deeply on the issues identified in the earlier version, and not just apply 'band-aid' solutions to the problems. The new manuscript now includes only terrestrial invertebrate assemblages, not aquatic assemblages, and is much more transparent about how the data have been analysed. The most problematic analyses from the earlier version have been dropped, and the revised analyses have corrected the earlier problems with the metrics used (most notably with the correct evenness metric now used - which shows a very interesting result).

Most importantly, I like the new conceptual framing in figure 1 around three different scenarios for proportional or asymmetrical declines in rare vs common species, and what that would mean for different components of biodiversity. This presents a MUCH more compelling and cohesive story, which promises to draw together many disparate findings in the previous literature on abundance and richness trends.

I also quite like the way that the authors have dealt with partial evidence for the 'proportional decline' scenario versus the 'abundant species decline more' scenario in the manuscript. There is a nice balance and gravitas in the considered approach to the discussion, and the progressive deployment of supplementary tests to (try to) tease apart the different scenarios. Arguably, none of these tests is sufficient in its own right to resolve the issue, but that is explicitly the thread of the authors' arguments (and not surprising, given the limitations inherent in the data). I think this makes a valuable and wholly transparent contribution to the debate, exactly as it is laid out in the manuscript.

I worked through the authors' responses to my earlier comments carefully, and agree that they have taken the appropriate actions in each case.

Overall, then, I was pleasantly surprised at the improvement in quality, and can readily see this an important advance in the field.

I have just a couple of minor queries.

In Figure 3b I was not really sure how readers were expected to interpret the double-headed arrow symbols on each SAD bar? Are these just conceptual/qualitative indicators (but then what relative value do they convey across each bar?), or are they meant to be quantitative in some sense?

In Figure 4 I was initially sceptical of what the correction for regression-to-mean effects might entail (and how it would be possible to do this analytically).

I think it is debatable how best to deal with adjustments for potential 'regression to mean' effects, such as those in the supplement, but I can't think of an alternative analytical approach to do it better. I appreciate that the authors have run sensitivity analyses to see how their conclusions might be altered by adjusting different parameter values, so perhaps it is OK? This might need an expert statistical opinion perhaps?

Referee #2 (Remarks to the Author):

After reviewing the original manuscript as reviewer 2, I have read the revision, reviewer reports and responses to the reviewers. Overall, my opinion about this article has not changed. The topic of research is highly relevant and I appreciate bringing together such a wealth of studies to provide an analysis of how insect biodiversity is in decline. My textual suggestions have been well addressed, and I feel that the choice of wording is now matching better what the manuscript is about. Nevertheless, I find that not much changed in how the manuscript extends beyond what other manuscripts have presented, especially noting that based on the same dataset similar results were presented by the authors before. This observation is shared by other reviewers that have seen the original version of the manuscript. In the original reviewer reports a list of similar studies is provided.

The revised version focusses more strongly on rare versus locally abundant species. To me, many of the results of earlier papers such as the study by Hallman on biomass decline already reveal that the extend of decline must include steep decline of locally abundant species. Although this point may have been made less explicit in these studies, I do not see how the current study is providing us with the significant new insights that I expect when reading Nature.

In the revised manuscript there are a number of aspect that I suggest to further revise. Some of these concern the conclusion of the manuscript.

Title: The declines of abundant species is not driving insect loss. It is characterizing insect loss. Drivers would be agricultural intensification, etc.

Line 30-36: The overall conclusion presented is that locally abundant species are declining, rare species not. In these lines of the abstract it is stated that also rare species are declining, but that these are replaced by other rare species. This is confusing and mismatches with the title and overall conclusion.

Line 45-48: Here the discussion of species interactions being the best metric in predicting consequences of biodiversity decline is completely left out.

Line 53-55: I do not agree that "knowing in particular the trends for rare or abundant species" is the metric to understand biodiversity decline. In particular it is understanding how interactions that are lost cause the patterns of decline.

Line 63: Here the authors acknowledge that we already know that at least the locally abundant species must be in decline given the rates of biodiversity decline.

Line 96: delete redundant space after evidence

Box 1: To me the box is misleading. The first figures that the readers will see are the theoretical ones in box 1. I find that these figures may be part of a supplementary to help the reader catch the expected patterns, but in the main document they may be confused for empirical data.

Line 138: Why are colonizing species considered to be mostly rare? How are invasive species considered here?

Line 140: The reference to the code should be included at the XXX

Line 181: It is unclear to me what the paragraph aims to illustrate. Do you mean to say that the narrative of insect decline is not true? What is the exact nuance that this paragraph aims to make?

Referee #3 (Remarks to the Author):

This revised draft of the manuscript is greatly improved, addressing most of my serious concerns, and those of other reviewers. I am now happy to endorse the paper's publication. I still have some (mostly relatively minor) concerns, which I would ideally like to see addressed in the final published version:

(a) The loss of some very rare spp and their replacement by others (e.g. lines 32-33) may reflect "pseudo-turnover" (JJ Beck et al, 2018, *Oikos* 271:1605-18) of species so rare that they are only occasionally sampled. The results text (lines 197-202) doesn't make it clear to what extent the patterns is that initially rare spp are not declining, or alternatively whether other (initially commoner) spp are becoming rare. However, the new population trend sections help answer these concerns.

(b) Are the scenarios described in Box 1 (and Fig 1) explicitly modeled, or are these subjective descriptions of the authors' expectations? I would have thought, for instance, that proportionate declines scenario [Fig 1(a) left panel] would have parallel lines (in log space) for even the rarest spp, but with them flat-lining when they decline to <1 individual. The SAD predictions [Fig.1 (b)] are poorly explained; it would be easier to accept them if they emerged from an explicit methodology.

(c) The fact that the authors (lines 157-8) "found much stronger declines in abundance than richness" is taken as evidence that "changes beyond a mere sampling process are occurring." However, a "mere sampling process" would predict precisely this pattern: typically, cutting the size of a sample by ca. 90% reduces species richness by about 50% (dubbed Darlington's rule).

Beyond those, there are some minor edits needed in places (e.g. line 48: replace "overseen" with "overlooked"; line 140: give the location for the code rather than "XXX"; line 144 and following: give time frame (per year?)), but overall the manuscript is clear and well-written.

Many thanks to the authors for the hard work they've done in revising the paper. The result was worth it.

Author Rebuttals to First Revision:

Detailed account of the changes made to data and analysis. The line numbers refer to the tracked changes version of the manuscript:

We have changed some parts of our statistical analysis of population trends (Fig 4):

- We now use Poisson distributed models to test the mean trends of the populations of the different abundance classes. This is more appropriate than our previous $\log(N+1)$ transformation, because especially for the rare species, the +1 transformation distorts the trends. This is now detailed in Methods L753-756

While this change does not change the results qualitatively, it has changed the estimates and the uncertainty (made them more negative), including that of the random effects (the dots in Fig 4) see lines 245-247: (-5.18-> 7.72% and 4.63% and -6.14 -> -0.64% and -4.05%)

- We no longer include the models of the species that were not present at the start of the time series, since all of these would start with $-\infty$. Some of those models did not converge after >10 days. By definition, the mean population trend of those initially absent species must be positive, and so losing this model does not alter our conclusions. We thus also removed the text about the species with the strongest increases, and explain this in L561-563 in the Methods
- When we re-ran the models, one unfortunately, did not converge for unclear reasons (abundance interval 2- 20-40%, classified on year 1, see fig 4). We solved this by excluding one small dataset that had a lot of zeroes, and removed the location level random effect (a level between the site and study level, that was meant to take spatial auto-correlation into account). We tested the effect of this exclusion on two other abundance intervals (80-100% and 60-80%). The difference in estimate was 3% and 10% of the CI respectively, and thus fell comfortably within the credible interval of the full model. We therefore believe this adjusted model structure is appropriate. This is explained in L760-766).
An alternative solution is to use one of the other abundance-classification approaches (years 1-2 or 1-5), which we present in EDfig 7.

Data used:

- We explain that we had to make some corrections to the underlying data in comparison to our previous work based on some recent scrutiny of the dataset (L84-85).
- This has slightly changed the number of sites available for analysis: 106 datasets and 923 sites) (lines 89-90 and Methods L483) Also see EDtable1, 2 and 3
- A few estimates of the biodiversity models have changed slightly after adjustments in the underlying data (L160-169), but this is negligible (mostly less than 2 decimals).
- The Result of EDfig8 has changed slightly (a less clear trend of steeper slopes of the random effects with increasing species abundance), and we have slightly rephrased our conclusions to indicate that the higher abundance classes have steeper trends than the lower abundance classes.

Other changes:

- We now refer to two recent papers on freshwater invertebrates in 2 places: L81-82 and L295-303 (and in line 470 in the Methods)
- We have added a legend for Fig 1
- We have deleted a paragraph as requested by reviewer 2. Part of the paragraph was included elsewhere (L178ff)
- Some more details on data availability: L550 and L691

Referee #1 (Remarks to the Author):

Review of MS 2022-06-08651A Nature

Widespread declines of formerly abundant species drive insect loss

van Klink et al.

20th June 2023

I commented on an earlier version of this manuscript (as reviewer #1) and had quite a number of reservations about the work. The resubmitted version is an entirely different beast. All credit () to the authors for taking the time to think deeply on the issues identified in the earlier version, and not just apply 'band-aid' solutions to the problems. The new manuscript now includes only terrestrial invertebrate assemblages, not aquatic assemblages, and is much more transparent about how the data have been analysed. The most problematic analyses from the earlier version have been dropped, and the revised analyses have corrected the earlier problems with the metrics used (most notably with the correct evenness metric now used - which shows a very interesting result).

Most importantly, I like the new conceptual framing in figure 1 around three different scenarios for proportional or asymmetrical declines in rare vs common species, and what that would mean for different components of biodiversity. This presents a MUCH more compelling and cohesive story, which promises to draw together many disparate findings in the previous literature on abundance and richness trends.

I also quite like the way that the authors have dealt with partial evidence for the 'proportional decline' scenario versus the 'abundant species decline more' scenario in the manuscript. There is a nice balance and gravitas in the considered approach to the discussion, and the progressive deployment of supplementary tests to (try to) tease apart the different scenarios. Arguably, none of these tests is sufficient in its own right to resolve the issue, but that is explicitly the thread of the authors' arguments (and not surprising, given the limitations inherent in the data). I think this makes a valuable and wholly transparent contribution to the debate, exactly as it is laid out in the manuscript.

I worked through the authors' responses to my earlier comments carefully, and agree that they have taken the appropriate actions in each case.

Overall, then, I was pleasantly surprised at the improvement in quality, and can readily see this an important advance in the field.

Response: Thanks for your kind words and support of our manuscript

I have just a couple of minor queries.

In Figure 3b I was not really sure how readers were expected to interpret the double-headed arrow symbols on each SAD bar? Are these just conceptual/qualitative indicators (but then what relative value do they convey across each bar?), or are they meant to be quantitative in some sense?

Response: These arrows were entirely conceptual and aimed to help the reader interpret the posterior distributions above. The reviewer's comment suggests to us that we did not succeed. We have removed this part of figure 3.

In Figure 4 I was initially sceptical of what the correction for regression-to-mean effects might entail (and how it would be possible to do this analytically).

I think it is debatable how best to deal with adjustments for potential 'regression to mean' effects, such as those in the supplement, but I can't think of an alternative analytical approach to do it better. I appreciate that the authors have run sensitivity analyses to see how their conclusions might be altered by adjusting different parameter values, so perhaps it is OK? This might need an expert statistical opinion perhaps?

Response: Thanks for the support. RtM is a major problem when analyzing data in relation to their starting values. We have done the best we could come up with to deal with this problem. One of our sensitivity analyses entirely removed any regression-to-mean effects by assigning the abundance intervals using the full time-series, not just the starting values. This analysis also found decreases of the most common species and more stable trends of the rare species (ED fig 7).

Referee #2 (Remarks to the Author):

After reviewing the original manuscript as reviewer 2, I have read the revision, reviewer reports and responses to the reviewers. Overall, my opinion about this article has not changed. The topic of research is highly relevant and I appreciate bringing together such a wealth of studies to provide an analysis of how insect biodiversity is in decline. My textual suggestions have been well addressed, and I feel that the choice of wording is now matching better what the manuscript is about. Nevertheless, I find that not much changed in how the manuscript extends beyond what other manuscripts have presented, especially noting that based on the same dataset similar results were presented by the authors before. This observation is shared by other reviewers that have seen the original version of the manuscript. In the original reviewer reports a list of similar studies is provided.

The revised version focusses more strongly on rare versus locally abundant species. To me, many of the results of earlier papers such as the study by Hallman on biomass decline already reveal that the extend of decline must include steep decline of locally abundant species. Although this point may have been made less explicit in these studies, I do not see how the current study is providing us with the significant new insights that I expect when reading Nature.

Response: We appreciate your comments and our findings may seem obvious to some specialists in the field. But, as we show in Fig. 1, overall declines in community biomass (e.g., as measured by

Hallman) can still arise when common species are declining at the same rate or even at a lesser rate as rare species. Hence, we think there is more important nuance that has been often overlooked.

In the revised manuscript there are a number of aspect that I suggest to further revise. Some of these concern the conclusion of the manuscript.

Title: The declines of abundant species is not driving insect loss. It is characterizing insect loss. Drivers would be agricultural intensification, etc.

Response: We have changed the title to: ' Disproportionate declines of formerly abundant species drive underly insect loss'

Line 30-36: The overall conclusion presented is that locally abundant species are declining, rare species not. In these lines of the abstract it is stated that also rare species are declining, but that these are replaced by other rare species. This is confusing and mismatches with the title and overall conclusion.

Response: We have changed the title to 'disproportionate' and 'underly'. The overall conclusion is that formerly abundant species, on average, decline more than formerly rare species. This is now better reflected in the title, abstract and overall conclusion.

Line 45-48: Here the discussion of species interactions being the best metric in predicting consequences of biodiversity decline is completely left out.

Response: We agree that species interactions are important and interesting to understand changes, but this is beyond the focus of our paper. In this paper, we focus on the question of 'what is changing?'. The questions 'why is it changing?' and 'what are the consequences of change?' are interesting, but not the focus of this paper.

We have removed 'and its consequences'.

Line 53-55: I do not agree that “knowing in particular the trends for rare or abundant species” is the metric to understand biodiversity decline. In particular it is understanding how interactions that are lost cause the patterns of decline.

Response: The reviewer is right that relative abundances are not the only way to look at biodiversity change, hence, we have removed the word 'only'.

Although beyond the scope of our study, from the perspective of ecosystem functioning and the consequences of insect loss, interaction loss is an important aspect to study.

Line 63: Here the authors acknowledge that we already know that at least the locally abundant species must be in decline given the rates of biodiversity decline.

Response: Correct, which is logical. The real question is thus whether the rare species decline more or less than the abundant ones. We now emphasize this more on lines 52-54.

Line 96: delete redundant space after evidence

Response: Done

Box 1: To me the box is misleading. The first figures that the readers will see are the theoretical ones in box 1. I find that these figures may be part of a supplementary to help the reader catch the expected patterns, but in the main document they may be confused for empirical data.

Response: In agreement with the editor, we will retain the box, but make clearer that these are conceptual figures, by adding the word 'conceptual' to the box title. We have added the methodology for making these figures to the Methods section, and provide all code.

Line 138: Why are colonizing species considered to be mostly rare? How are invasive species considered here?

Response: We focused on simulating different patterns of species decreases, since our previous analysis found evidence for a mean decrease in community-level insect abundance. Here, we make clearer that we did not include colonists or any sort of increasing species, including invasives. But we recognize that they would balance some of the loss in our toy model.

Line 140: The reference to the code should be included at the XXX

Response: We have added the link to our Zenodo repository

Line 181: It is unclear to me what the paragraph aims to illustrate. Do you mean to say that the narrative of insect decline is not true? What is the exact nuance that this paragraph aims to make?

Response: We see that the structure of this paragraph is confusing. . We have removed most of the paragraph and moved the sentences about our sensitivity analyses to the previous paragraph.

Referee #3 (Remarks to the Author):

This revised draft of the manuscript is greatly improved, addressing most of my serious concerns, and those of other reviewers. I am now happy to endorse the paper's publication. I still have some (mostly relatively minor) concerns, which I would ideally like to see addressed in the final published version:

(a) The loss of some very rare spp and their replacement by others (e.g. lines 32-33) may reflect "pseudo-turnover" (JJ Beck et al, 2018, *Oikos* 271:1605-18) of species so rare that they are only occasionally sampled. The results text (lines 197-202) doesn't make it clear to what extent the patterns is that initially rare spp are not declining, or alternatively whether other (initially commoner) spp are becoming rare. However, the new population trend sections help answer these concerns.

Response: This is an important point. Indeed, neither the SAD analysis, nor the biodiversity metrics can show this, which is why we have included the population analysis. By necessity, the trend in the number of rare species (Fig 3) is a combination of initially rare species staying rare, pseudo turnover (constantly very rare species popping in and out of the samples), previously abundant species

becoming rarer, and colonization by previously absent species. We now make clearer the limits of what this particular analysis can tell us on lines 218 ff; which provides more justification for our subsequent population analysis that aims to disentangle some of this complexity: 'Here it must be considered that the trend in this SAD interval represents unknown proportions of stable rare species, additions of previously more abundant species, local extinctions and/or newly colonizing species'. Unfortunately, there is no space to also include this in the abstract.

(b) Are the scenarios described in Box 1 (and Fig 1) explicitly modeled, or are these subjective descriptions of the authors' expectations? I would have thought, for instance, that proportionate declines scenario [Fig 1(a) left panel] would have parallel lines (in log space) for even the rarest spp, but with them flat-lining when they decline to <1 individual. The SAD predictions [Fig.1 (b)] are poorly explained; it would be easier to accept them if they emerged from an explicit methodology.

Response: These figures do stem from an explicit methodology, which we have now added to the methods section. For figure 1a, we have taken some freedom for the sake of clarity. We show non-integer values in the conceptual model (although species can only be counted in integers), because showing the declines in integer values leads to sharp steps in the lines, which we feel would be a distraction for a reader.

We have now included flat lining low abundances, but it's arbitrary from which abundance value (y axis) this should occur. We have now used 0.6 as cut-off, for illustrative purposes.

(c) The fact that the authors (lines 157-8) "found much stronger declines in abundance than richness" is taken as evidence that "changes beyond a mere sampling process are occurring." However, a "mere sampling process" would predict precisely this pattern: typically, cutting the size of a sample by ca. 90% reduces species richness by about 50% (dubbed Darlington's rule).

Response: The reviewer is of course right that changes in species richness must be weaker than changes in abundances, if only due to sampling fewer individuals. The 90%/50% rule is of course a rule of thumb and depends on the exact shape of the SAD. Therefore we have reworded this sentence as: "Although concomitant declines in abundance and species richness are expected from a mere sampling process (i.e. fewer species would be expected when fewer individuals are sampled from a species pool^{13,21}), this may still obscure changes to species' relative abundances, depending on the shape of the SAD."

Beyond those, there are some minor edits needed in places (e.g. line 48: replace "overseen" with "overlooked"; line 140:

Response: Done

give the location for the code rather than "XXX";

Response: Done

line 144 and following: give time frame (per year?)),

Response: This is an important oversight on our side. We have added 'per year'

but overall the manuscript is clear and well-written.

Response: Thank you

Many thanks to the authors for the hard work they've done in revising the paper. The result was worth it.